***Nat Hum Behav.* Author manuscript; available in PMC 2026 March 21.**

# Structural and functional asymmetry of the neonatal cerebral cortex

**Logan Z. J. Williams**[a,b,*], **Sean P. Fitzgibbon**[c], **Jelena Bozek**[d], **Anderson M. Winkler**[e], **Ralica Dimitrova**[a], **Tanya Poppe**[a], **Andreas Schuh**[f], **Antonios Makropoulos**[a], **John Cupitt**[f], **Jonathan O'Muircheartaigh**[a,g,h], **Eugene P. Duff**[c], **Lucilio Cordero-Grande**[a,i], **Anthony N. Price**[a], **Joseph V. Hajnal**[a], **Daniel Rueckert**[f,j], **Stephen M. Smith**[c], **A. David Edwards**[a,g,h], **Emma C. Robinson**[a,b,*]

[a]Centre for the Developing Brain, Department of Perinatal Imaging and Health, School of Biomedical Engineering and Imaging Sciences, King's College London, London, SE1 7EH, UK.

[b]Department of Biomedical Engineering, School of Biomedical Engineering and Imaging Science, King's College London, London, SE1 7EH, UK.

[c]Centre for Functional MRI of the Brain (FMRIB), Wellcome Centre for Integrative Neuroimaging, Nuffield Department of Clinical Neurosciences, University of Oxford. John Radcliffe Hospital, Oxford, OX3 9DU, UK.

[d]Faculty of Electrical Engineering and Computing, University of Zagreb, Zagreb, Croatia.

[e]Emotion and Development Branch, National Institute of Mental Health, National Institutes of Health, Bethesda, Maryland, USA.

[f]Department of Computing, Imperial College London, London SW7 2RH, UK.

[g]Department for Forensic and Neurodevelopmental Sciences, Institute of Psychiatry, Psychology and Neuroscience, King's College London, London, SE5 8AF, UK.

[h]MRC Centre for Neurodevelopmental Disorders, King's College London, London, SE1 1UL, UK.

[i]Biomedical Image Technologies, ETSI, Telecomunicación, Universidad Politécnica de Madrid and CIBER-BBN, Madrid, 28040, Spain.

[j]Klinikum rechts der Isar, Technical University of Munich, Munich, Germany.

## Abstract

Features of brain asymmetry have been implicated in a broad range of cognitive processes; however, their origins are still poorly understood. Using a new left-right symmetric,

*Corresponding author, Correspondence: logan.williams@kcl.ac.uk; emma.robinson@kcl.ac.uk.

**Author Contributions**
L.Z.J.W., S.P.F., A.M.W., T.P., J.O., E.P.D., S.M.S., A.D.E. and E.C.R. designed the research, L.Z.J.W., S.P.F., J.B., A.S., A.M. and J.C. performed research, L.Z.J.W. analyzed data, L.Z.J.W., A.D.E. and E.C.R. wrote the manuscript. All authors reviewed and edited the manuscript.

**Competing Interests**
The authors declare that they have no competing interests.

spatiotemporal cortical surface atlas, we investigated cortical asymmetries in 442 healthy term-born neonates using structural and functional magnetic resonance images from the Developing Human Connectome Project. Cortical asymmetries observed in the term cohort were contextualised in two ways: by comparing them against cortical asymmetries observed in 103 preterm neonates scanned at term-equivalent age, and by comparing structural asymmetries against those observed in 1110 healthy young adults from the Human Connectome Project. Our results demonstrate that the neonatal cortex is markedly asymmetric in both structure and function, and while associations with preterm birth and biological sex were minimal, significant differences exist between birth and adulthood. Although these changes may represent experience-dependent developmental milestones, longitudinal studies across the lifespan are required to fully address these hypotheses.

**Keywords**

developing human connectome project; cortical surface; asymmetry; resting-state networks; preterm birth

## Introduction

Asymmetry is an important feature of human brain organisation (1), implicated in a number of cognitive processes including language (2, 3) and face processing (4). Changes to typical brain asymmetry have been associated with neurodevelopmental and psychiatric disorders including autism spectrum disorder (5, 6), schizophrenia (7, 8), obsessive-compulsive disorder (9), attention-deficit hyperactivity disorder (10), and dyslexia (11). However, the origins of these asymmetries (12) and the factors affecting their development (13) remain poorly understood.

Using magnetic resonance imaging (MRI) to characterise cortical asymmetries in the neonatal period can help clarify their ontogenesis (14). For example, one of the most well-characterised structural asymmetries of the human cerebral cortex is the right-deeper-than-left asymmetry of the superior temporal sulcus, which has been demonstrated in fetuses (15, 16), preterm (17, 18) and term neonates (19, 20), children and adults (19, 21, 22). Other sulcal depth asymmetries that have been described include the rightward asymmetry of the parieto-occipital sulcus in neonates (20), and adults (19), and the leftward asymmetry of the lateral fissure (18–20).

In contrast, asymmetries of surface area and cortical thickness during the neonatal period have received less attention. Li et al. demonstrated leftward surface area asymmetries in the precentral, postcentral, middle frontal and cingulate gyri, temporal pole and insula; as well as rightward surface area asymmetry in the inferior parietal, posterior temporal and lateral occipital cortices (20); and rightward cortical thickness asymmetries in the temporal and lateral occipital cortices, and insula (23). Functional asymmetries, measured using task-based functional MRI (fMRI), are also present at birth and during infancy, and include lateralised responses to speech (24, 25) and music (26).

Although biological sex has been associated with numerous measures of neonatal brain structure and function (27), studies investigating associations between cortical asymmetries and biological sex are limited. Dubois et al. found an association between biological sex and the emergence of sulci, with sulci on the right hemisphere emerging significantly earlier than the left in biological females, but not males (17). However, Li et al. found no association between biological sex and cortical thickness asymmetries in a longitudinal cohort of neonates scanned at birth, 1 and 2 years of age (23). Relationships between postmenstrual age (PMA) and cortical asymmetries in neonates have received even less attention, with Dubois et al. reporting no association between PMA and sulcal depth asymmetry in preterm neonates born between 26 - 36 weeks PMA (17). To our knowledge, no studies have investigated how biological sex or PMA are associated with functional asymmetry in the neonatal period.

While these studies have greatly advanced our understanding of brain asymmetry in early life, they also have important limitations. Studies that compare cortical features based on volumetric registration (24, 26, 28–30) fail to account for structural and functional heterogeneity of the cortex, and can lose statistical power due to residual misalignment between cortical folds during registration (31, 32). In the case of fMRI, this may also lead to leakage of blood-oxygen-level-dependent signal across adjacent cortical folds (33). Likewise, previous surface-based analysis of neonatal asymmetries have been performed on a single population average template of brain folding; whereas studies have repeatedly shown that correspondence between subjects may be improved through use of age-specific templates that better capture the dynamism of cortical development over the perinatal period (34–36).

## Results

The Developing Human Connectome Project (dHCP) is an open-science project that has advanced image acquisition and processing to produce a large cross-sectional set of multi-modal MRI data in neonates between 24-45 weeks PMA. Using 887 scan sessions from 783 neonates (578 healthy term-born), we developed a new left-right symmetric, surface-based spatiotemporal atlas that models population average changes in cortical morphology between 28-44 weeks PMA. Following surface-based registration to this new template, we investigated structural asymmetry in 442 healthy term-born neonates at a vertex-wise level using permutation testing (37). As there are only a limited number of studies directly testing for functional asymmetry in the neonatal period (38), we also investigated asymmetries of intrinsic cortical activity using resting-state fMRI (rs-fMRI) mapped to the cortical surface, and tested for associations between cortical asymmetries, biological sex and PMA. Key demographic variables are reported in Table 1 and Figure 1. Importantly, right hemispheric volume was significantly greater than the left, with a hemispheric volume asymmetry (left - right) of $-1.8 \text{cm}^3$ ($-0.9$ - $-2.8$, $p < 0.001$).

### Structural Asymmetries in the Healthy Term Neonatal Brain

Figure 2 and Figure 3 show asymmetry indices and $-\log_{10}(p)_{mcfwe}$ value maps, respectively, for sulcal depth, surface area and cortical thickness. The sulcal depth asymmetry index

map demonstrated an interesting centre-surround pattern, where regions of asymmetry in one direction were flanked by regions of opposing asymmetry, for example a centre of rightward asymmetry in the supramarginal gyrus surrounded by regions of leftward asymmetry. However, only a limited number of sulci demonstrated significant asymmetry $(-\log_{10}(p)_{mcfwe} > 1.3)$ after registration to a fully symmetric template (Methods: Creating Left-Right Symmetric Surface Atlas). Leftward asymmetry (left deeper than right) was observed for the anterior and posterior lateral fissure. There was also rightward asymmetry (right deeper than left) of the middle portion of the superior temporal sulcus.

Regarding surface area, there were leftward asymmetries (left > right) located in the supramarginal, medial precentral, postcentral, posterior cingulate, caudal anterior cingulate, caudal superior frontal and inferior temporal gyri, superior lateral occipital lobe and anterior insula. There were rightward asymmetries (right > left) located in the superior and middle temporal, lingual, rostral anterior cingulate, rostral superior frontal, middle frontal, and inferior frontal gyri, insula and parieto-occipital sulcus.

Compared to the asymmetry indices for sulcal depth and surface area, cortical thickness asymmetry indices were smaller in magnitude (Figure 2). There were leftward cortical thickness asymmetries (left > right) located in the supramarginal, angular, precentral, postcentral, superior frontal, and inferior temporal gyri, lateral occipital lobe and insula. There were also rightward asymmetries (right > left) located in the superior temporal, middle temporal, caudal superior frontal and anterior cingulate gyri, orbitofrontal cortex, precuneus, and calcarine and parieto-occipital sulci. As a reference, asymmetry indices for these structural measures are also reported on a per-region basis (Figure S1).

### Functional Asymmetry in the Healthy Term Neonatal Brain

Group independent component analysis (ICA) run on the cortical surface generated 16 resting-state networks which are shown in Figure S2. These include primary (somatosensory, medial and lateral motor, auditory, medial and lateral visual) and association (rostral and caudal prefrontal, motor association, medial and lateral parietal, temporoparietal and cingulo-opercular) networks. The asymmetry indices of these networks were all within approximately the same range, and were overall similar to the asymmetry indices for sulcal depth and surface area (Figure 2). Significant asymmetries were found in 11 of these networks (Figure 3).

**Auditory Networks:** Asymmetry in the auditory network comprised a leftward asymmetric region located along the anterior end of the superior temporal gyrus. The temporoparietal network demonstrated a small leftward asymmetric region centred on the middle superior temporal sulcus.

**Visual Networks:** The primary and association visual cortices demonstrated marked asymmetry. The medial visual network consisted of a single rightward asymmetric region located on the medial aspect of the occipital lobe. Rightward asymmetries in the dorsal visual association network were located in the intraparietal sulcus and lateral occipital lobe, whilst rightward asymmetries in the ventral visual association network were observed in the inferior occipital gyrus and part of the posterior fusiform gyrus.

**Sensorimotor Networks:** In the medial motor network, a leftward asymmetric region was located on the dorsal aspect of the central sulcus, and a rightward asymmetric region on the dorsomedial aspect of the central sulcus. The lateral motor network contained a leftward asymmetric region located in the midportion of the central sulcus and a rightward asymmetric region located on the inferolateral aspect of the central sulcus and dorsal mid-insula. For the somatosensory network, the leftward asymmetric region was located in the midportion of the postcentral gyrus, and the rightward asymmetric region on the inferolateral aspect of the postcentral gyrus.

**(Proto) Default Mode and Control Networks:** The precuneus and lateral parietal networks demonstrated rightward asymmetry, with asymmetry in the precuneus localised to the subparietal sulcus and posterior cingulate gyrus, and asymmetry in lateral parietal network located on the intraparietal sulcus. The cingulo-opercular network demonstrated rightward asymmetry centred on the caudal anterior cingulate gyrus.

### Postmenstrual Age and Biological Sex

Associations between PMA and cortical asymmetries were minimal, with surface area asymmetry in the anterior insula decreasing linearly with PMA, and surface area of a smaller region in the mid-insula increasing linearly with PMA (Figure S3). Figure S4 shows the difference in asymmetry indices between biological females and males (female - male) for all cortical measures. No statistically significant associations between biological sex and structural or functional cortical asymmetries were observed (Figure S5).

### Preterm Birth

It is well known that neurodevelopment in preterm neonates (born < 37 weeks gestational age (GA)) is characterised by dysmaturation (39, 40), and that those born preterm have poorer long-term developmental and behavioural outcomes compared to their term-born counterparts (41–44). By comparing cortical asymmetries between the same 442 term-born neonates and 103 preterm neonates at term-equivalent age (TEA), we directly tested for associations between preterm birth and the development of structural and functional asymmetry in the newborn cerebral cortex. Baseline demographic variables of the preterm cohort are shown in Table 1 and Figure 1. As for the term cohort, the right hemisphere in preterm neonates at TEA was significantly larger than the left, with a hemispheric volume asymmetry of $-1.9\text{cm}^3$ ($-0.9$ - $-3.0$, $p < 0.001$).

Figure S6 shows the difference in asymmetry indices between term and preterm-born neonates (term - preterm) for all cortical measures. Only one very small region in the anterior insula exhibited a significant difference in cortical thickness asymmetry between the term and preterm cohorts scanned at TEA (term > preterm). No significant differences between term and preterm neonates were observed in the other structural (sulcal depth and surface area) or functional cortical asymmetries (Figure S7).

### Structural Asymmetry Changes Between Birth and Young Adulthood

Finally, despite the number of studies assessing the trajectory of cortical asymmetries across the lifespan (45–47), there is little knowledge of how the cortical asymmetries established

at birth differ from those seen in later life. To address this, we compared structural cortical asymmetries seen in the term neonatal cohort with those in 1110 healthy young adults from the Human Connectome Project - Young Adult (HCP-YA). Participants had a median (interquartile range (IQR)) age at scan of 29 (26 to 32) years, and total brain volume of 999.0 cm$^3$ (925.5 to 108.0), and median hemispheric volume asymmetry of $-7.9$cm$^3$ ($-5.0$ to $-10.6$, $p < 0.001$).

Figure 4a demonstrates the asymmetry indices of structural asymmetries in the HCP-YA cohort. The HCP-YA sulcal depth asymmetry index map also demonstrated the same centre-surround pattern. Significant sulcal depth asymmetries were observed for the lateral fissure (leftward), and superior temporal and parieto-occipital sulci (rightward). Leftward surface area asymmetries were observed in the lateral postcentral, inferior temporal and superior frontal gyri, lateral and medial occipital lobe, and superior temporal sulcus. Rightward surface area asymmetries were observed in the superior temporal, middle temporal and cingulate gyri, frontal lobe and insula. Cortical thickness asymmetry was largely rightward asymmetric, with several regions of leftward asymmetry localised to the precentral, postcentral, inferior temporal and parahippocampal gyri, and cingulate and calcarine sulci (Figure 4c). Region-of-interest summaries of these structural metrics are provided in Figure S1. There were no linear associations with age (Figure 4e) but biological sex was significantly associated with cortical thickness asymmetry, with biological females having increased leftward asymmetry in the posterior middle and inferior temporal gyri, and lateral occipital lobe compared to biological males (Figure 4d).

Figure 4b also summarises previously published results from Sha et al. (see Methods: Data Availability), who performed vertex-wise cortical thickness and surface area asymmetry analysis for 31,864 data sets from the UK Biobank (UKB) (48–50). These asymmetry maps are smoother than those from our own analyses, as they are calculated on lower resolution surfaces (10,252 vertices per hemisphere, compared to 32,492 vertices per hemisphere used here) (48). Leftward surface area asymmetries were observed in the postcentral, supramarginal and superior frontal gyri. Rightward surface area asymmetries were observed in the posterior temporal, lateral occipital and inferior parietal lobes. For cortical thickness, leftward asymmetries were observed for the postcentral gyrus, frontal lobe and cingulate sulcus. Rightward cortical thickness asymmetries were seen in the superior temporal, middle temporal and supramarginal gyri, lateral occipital and inferior parietal lobes, and parieto-occipital sulcus.

Finally, Figure 5 displays results comparing structural asymmetries between neonates and adults. The magnitude of asymmetry indices were similar between both populations (Figure 5a and Figure 5b). Young adults had higher surface area asymmetry indices (left > right) compared to neonates along the length of the superior temporal sulcus/gyrus, and superior portion of the parieto-occipital sulcus. In contrast, neonates had higher cortical thickness asymmetry indices (left > right) in the supramarginal and lingual gyri, lateral occipital, inferior parietal, and temporal cortices, and anterior and posterior insula. No differences in sulcal depth asymmetries were observed between these two groups (Figure 5c).

## Discussion

This paper addresses a number of outstanding questions concerning the development of cortical asymmetry in the neonatal period, including associations with biological sex, PMA and preterm birth. It also directly assesses how structural cortical asymmetries differ between birth and young adulthood using two large-scale, state-of-the-art, publicly available datasets. In all cases, analysis was contingent on accurate correspondence of features between hemispheres and across individuals, which was achieved through: 1) development of a novel left-right symmetric, spatiotemporal cortical surface atlas (based on work described in Bozek et al. (36) and Garcia et al. (51)), and 2) biomechanically-constrained surface registration to this template using Multimodal Surface Matching (MSM) (52, 53) and template-to-template registration (34–36). Overall, our findings highlight the sensitivity of surface-based analysis methods for detecting both structural and functional cortical asymmetries, which have been previously underestimated. They also underscore the advances in image acquisition and processing that have been made as part of large-scale neuroimaging consortia.

Our results not only replicate findings reported in earlier neonatal studies, but also identify new asymmetries that have previously been observed in older cohorts. The rightward depth asymmetry of the superior temporal sulcus and leftward depth asymmetry of the lateral fissure presented here are in agreement with previous studies in preterm neonates both before and at TEA (18). Of the surface area asymmetries reported by Li et al., we replicate leftward asymmetry of the postcentral and caudal cingulate gyri, temporal pole and anterior insula, and rightward asymmetry in the inferior parietal, posterior temporal and inferior lateral occipital cortices (20). We also replicate cortical thickness asymmetries observed in the superior and middle temporal gyri (23).

Our results further build upon earlier studies by identifying novel asymmetries in both surface area and cortical thickness. Regarding surface area, we observed leftward asymmetry in the superior lateral occipital lobe, and rightward asymmetry spanning the superior frontal, middle frontal, inferior frontal and rostral anterior cingulate gyri, medial occipital lobe, and mid-insula. For cortical thickness, we found leftward asymmetry of the inferior temporal, supramarginal, angular, precentral and postcentral gyri, lateral occipital lobe and insula, and rightward asymmetry of the caudal superior frontal and anterior cingulate gyri, orbitofrontal cortex, precuneus, and calcarine and parieto-occipital sulci. Some of these newly recognised asymmetries have been described in older cohorts. For instance, Zhou et al. demonstrated rightward asymmetry of cortical thickness in the inferior frontal gyrus and medial occipital lobe, and leftward asymmetry of cortical thickness in the supramarginal and angular gyri, and lateral occipital lobe in children and adolescents (46). More recently, Roe et al. demonstrated rightward surface area asymmetry of the medial occipital lobe and caudal anterior cingulate gyrus from as early as 4 years of age (47).

Although functional asymmetries have not previously been investigated using surface-based analyses, our findings are consistent with volumetric-based studies in neonates and adults. Studies investigating language using speech stimuli in task-based fMRI have shown leftward asymmetric activation along the peri-Sylvian region in neonates (24, 25), children (54) and

adults (2, 3). The leftward asymmetry of the auditory network seen here is also consistent with leftward asymmetries in auditory networks observed in children (54) and adults (55), and might be related to a leftward asymmetric increase in fractional anisotropy of the arcuate fasciculus that has been observed in neonates (29, 56) and adults (57, 58). Additionally, we identified a region of leftward asymmetry in the temporoparietal network centred on the middle portion of the superior temporal sulcus. Together, the asymmetries of the auditory and temporoparietal networks share the same anatomical landmarks as the anterior and middle temporal voice areas, respectively (59). These areas constitute part of a functional network responsible for processing voice (59, 60) that appears to be closely associated with the rightward depth asymmetry of the superior temporal sulcus (59, 61). The presence of this rightward superior temporal sulcus asymmetry in humans across the lifespan (in both health and disease), but not in non-human primates, suggests that this asymmetry might be related to the evolution of human-specific cognitive abilities such as language (21). Moreover, the depth asymmetry of the superior temporal sulcus has been associated with two genes (*DACT1* and *DAAM1*), which are mainly expressed during prenatal brain development (62). These findings lend further support to the importance of cortical asymmetries in the peri-Sylvian region for language development in early life.

The marked asymmetries observed in the neonatal visual networks resemble those seen in adults. Specifically, our findings demonstrate rightward asymmetry of the dorsal visual association network along the intraparietal sulcus, which has been implicated in spatial attention (63). Additionally, there was a small region of rightward asymmetry in the lateral occipital lobe, which contains cortical areas involved in biological motion perception in infants (64) and adults (65). We also observed rightward asymmetry in the ventral visual association network, localised to the inferior occipital gyrus and part of the fusiform gyrus, which is consistent with rightward asymmetric electroencephalographic responses to face-like stimuli in the occipitotemporal cortex of neonates (66) and infants (67). Furthermore, it is well established that there are right-hemispheric preferences for spatial attention, biological motion perception, and face processing in adults (68). Combined, these findings suggests that functional asymmetries in visual association networks may underlie the right hemispheric specialisation of spatial attention, biological motion and face processing seen in later life.

In addition to recapitulating well-documented functional asymmetries in the auditory and visual systems seen in adulthood, we observed a pattern of functional asymmetry in the somatomotor networks that might reflect the somatotopic organisation of the sensorimotor cortex (69–71). Namely, the somatosensory and lateral motor networks showed leftward asymmetry in the upper limb region of the somatotopic map (69). This functional asymmetry might be associated with microstructural asymmetry (increased fractional anisotropy) of the left corticospinal tract compared to the right, which has previously been demonstrated in neonates (29), adolescents (72) and adults (73). It has also been shown that leftward lateralisation of motor network connectivity was associated with better motor performance in children (74), and that motor circuit connectivity in children diagnosed with autism spectrum disorder was more rightward asymmetric compared to children without autism spectrum disorder (75). This asymmetry may also relate to handedness, as genomic loci

associated with handedness (76, 77) are also associated with brain asymmetry and are expressed during prenatal brain development (78).

The somatosensory and lateral motor networks also demonstrated rightward asymmetry in the mouth region in the somatomotor homunculus which, in neonates, includes the mid-dorsal insula (69). We hypothesise that this rightward asymmetry is important for the development of swallowing in the neonatal period. Swallowing is a highly complex motor function that is represented across a number of cortical areas including the primary somatomotor cortices (79) and mid-insula (80, 81). Moreover, swallowing is initiated in response to a number of stimuli (79), two of which (taste and interoception), are represented in the right mid-dorsal insula (82). Together, this rightward asymmetry seen in neonates could reflect integration of sensorimotor, taste and interoceptive information that is critical for the development of swallowing.

Not all of the networks identified during the neonatal period demonstrated significant functional asymmetry. In particular, few association networks were asymmetric, which is consistent with previous work showing that the maturation of resting state networks in neonates follows a sequence from primary to association areas (83–86). Moreover, as association networks have been implicated with higher-order cognitive processes in adults, the emergence of asymmetries in these networks may be more experience-dependent. For instance, functional asymmetries in the frontal cortex have been associated with emotional regulation (87, 88), reward responses (89) and cognitive set shifting (the process of consciously redirecting attention) (90). Further research is required to establish how functional asymmetries at birth relate to cognition and behaviour in later life.

In contrast to earlier work (17, 20), we observed no associations between biological sex and structural or functional asymmetries in the neonatal period. Recent work by Williams et al. demonstrated that both total brain volume and hemispheric volume asymmetry were important explanatory variables when investigating differences in cortical asymmetries between biological sexes (91). The absence of these covariates in previous analyses may be one explanation of why the results presented here differ. Our results also suggest that observed sex-related differences in cortical asymmetry (1, 21, 91) likely emerge in later life. Further research is required to address when and how these sex-related differences in asymmetries emerge.

Associations between PMA and cortical asymmetries were minimal, with surface area asymmetry in the anterior insula decreasing with age, and surface area asymmetry in the midinsula increasing with age. The relative lack of age-related associations with cortical asymmetries is consistent with previous studies (17, 23), but also striking given the highly dynamic nature of cortical development over this period. One limitation of the current analysis is that it only tests for linear associations. As non-linear age associations with cortical asymmetries have been recently demonstrated from 4 - 89 years of age (47), future work should investigate whether similar trends exist in the perinatal period.

Given the marked impact that preterm birth has on cortical morphology (92–94) and functional connectivity (84, 86, 95, 96), it was surprising to find minimal differences in

cortical asymmetry between preterm and term-born neonates at TEA. This is in contrast to Kwon et al., who found differences in functional asymmetry between preterm and term-born neonates at TEA (38). However, that study was limited by relatively small sample sizes (38). Moreover, previous work has shown that structural asymmetries in preterm (97) and term (19, 20) neonates are similar, and recently, Liu et al. reported no differences in white matter asymmetry between preterm and term-born neonates at TEA (98). Together, these findings indicate that brain asymmetries may be robust to the effects of preterm birth, and that "multiple hits" are required to disrupt brain asymmetry (for example, preterm birth *and* polygenic risk) (99, 100). This may explain why cortical asymmetry is altered in psychiatric and neurodevelopmental disorders like schizophrenia (7, 8) and autism spectrum disorder (5, 6), which have complex and multifactorial aetiologies (101, 102). An important caveat is that we did not investigate how the degree of preterm birth affected cortical asymmetries. Decreasing GA at birth (especially <28 weeks) is associated with a higher incidence and severity of neurodevelopmental sequelae (41, 44), but our preterm cohort largely consisted of neonates born moderate-to-late preterm (born between $32^{+0}$ - $36^{+6}$ weeks) (see Figure 1 and Table 1). This might also explain why our findings differ from Kwon et al. (38).

Assessing structural asymmetry within the HCP-YA was performed as an initial step towards investigating differences between neonates and adults. We also qualitatively compared these results against mean structural asymmetry maps calculated in 31,864 UKB participants (48). Although these average maps bear some resemblance to our own results, they also demonstrate important differences including 1) leftward surface area asymmetry in medial occipital lobe in HCP-YA, but rightward in UKB and 2) rightward asymmetry in frontal lobe in HCP-YA but leftward in UKB. This is perhaps unsurprising as a number of important differences exist between this study and ours in terms of image acquisition, processing and analysis. Specifically, the input data in this study is of higher resolution, which is important as Glasser et al. demonstrated the benefit of reducing voxel size from 1mm isotropic to 0.7mm isotropic on the accuracy of cortical thickness measurements, especially in areas such as the medial occipital lobe where cortical ribbon is thinner (103). We also performed statistical analysis on higher resolution surfaces (32,492 compared to 10,252 vertices), which likely explains the higher resolution features of the asymmetry analysis in Figure 4 relative to Sha et al. (48). Although Roe et al. reported high correlation of vertex-wise structural asymmetries between the HCP-YA and UKB on higher resolution cortical surfaces (47), surface extraction using only T1-weighted (T1w) images (rather than both T1w and T2-weighted (T2w) images) and high spatial smoothing were image processing decisions that did not fully capitalise on the quality of the HCP-YA data (103).

Unlike sulcal depth, the direction of surface area and cortical thickness asymmetries significantly differed between birth and adulthood. These asymmetry reversals might represent neurodevelopmental milestones. Shaw et al. demonstrated that the pattern of cortical thickness asymmetries observed in the orbitofrontal (left > right) and lateral occipital (right > left) cortices during childhood reversed by adulthood. Moreover, the reversal in orbitofrontal cortical thickness asymmetry between childhood and adulthood was disrupted in those with attention-deficit hyperactivity disorder (104). Previous longitudinal studies assessing cortical asymmetries (45–47, 104) have not considered the perinatal period, which is one of the most dynamic phases of brain development (105). As our

results are cross-sectional, longitudinal studies mapping cortical asymmetries from the perinatal period onwards will be important for characterising asymmetry reversals in typical development, and how they relate to the acquisition of new behaviours.

Such studies may also shed light on how these asymmetries emerge. For example, it has been suggested that cortical thickness asymmetries are, in part, experience-dependent and arise through processes such as intracortical myelination (47) (which is inversely correlated with cortical thickness (106)). On the other hand, surface area asymmetries are thought to reflect the organisation of cortical minicolumns established prenatally (47), with postnatal changes potentially occurring as a result of genetically-programmed developmental processes (107), such as apoptotic cell death (108).

The similarity of sulcal depth and surface area asymmetries between neonates and adults suggest that these are broadly conserved, and may explain why the asymmetry indices of these measures are higher than the asymmetry indices of cortical thickness seen here and elsewhere (47, 48). It is possible that such conservation is, at least in part, genetically determined. Recent work investigating the genomic loci underpinning structural brain asymmetries has implicated genes that are expressed during prenatal brain development (78), which are involved in molecular pathways responsible for left-right axis patterning in vertebrates (78), visual and auditory pathway development (109), neural activity during development of the somatosensory cortex (110), and regulation of axonal guidance and synaptogenesis (111). There is also evidence that gene expression during prenatal brain development is asymmetric (112). Whilst genetic factors likely play a role in driving cortical asymmetries, they do not fully account for why these asymmetries, particularly functional asymmetries, are present at birth. There is emerging evidence suggesting that structural and functional connectivity preceding task-based function may represent a general mechanism of cortical development (113–115). Our results could represent an extension of this mechanism, whereby asymmetric functional connectivity precedes asymmetric task-based function in the auditory, visual and somatomotor networks.

The sulcal depth asymmetry index maps across both dHCP (Figure 2) and HCP-YA (Figure 4a) cohorts demonstrated a marked centre-surround pattern. It is difficult to fully dis-entangle the causes of this phenomenon but it is likely to be a combination of true differences in sulcal depth asymmetry and unavoidable residual image registration misalignment (Figure S13). Despite driving registration to a template that is perfectly symmetric (Figure 6), it is not possible for a template to capture all possible variations in cortical folding (116–118). Moreover, asymmetric presence of sulci has been demonstrated in a number of cortical regions including the (para)cingulate cortex (119, 120) and the frontal operculum (121), suggesting that in some instances folding correspondence between hemispheres within subjects is also not possible. This anatomical variability is compounded by the fact that image registration is an ill-posed problem with many possible solutions, where regularisations are introduced to enforce solutions that are considered biologically 'plausible'. It is possible that the effect of regularisation is different for the left and right hemispheres. Given the above, it is not possible to tell with certainty which centre-surround patterns are true biological differences or a product of residual misalignment. Regardless,

very few of these patterns were statistically significant, and for the purpose of the analyses presented here, should be seen as background noise.

An important methodological consideration in this study was the modality driving surface registration. Cortical surfaces were registered to a symmetric template using sulcal depth maps, which may have impacted the investigation of functional asymmetry, particularly in cortical regions where structural and functional correspondence is poor (122). An appealing alternative was to drive registration in a multimodal manner, and assess both structural and functional asymmetries using the HCP_MMP1.0 atlas (122). However, as the intersubject correspondence of secondary and tertiary sulci is relatively poor following multimodal alignment (32, 122), such an approach would have limited the ability to detect sulcal depth asymmetries compared to the approach used here (Figure S8). Moreover, multimodal registration of neonatal cortical surfaces to the HCP_MMP1.0 template assumes that registration may be driven with the same cortical features (resting state networks, T1w/T2w ratio and visuotopic maps) as used for adults, which requires that they are present and have comparable topography in neonates. However, current evidence suggests that this is not the case (123, 124). Ultimately, multimodal registration of neonates to the adult HCP_MMP1.0 template is non-trivial, and outside the scope of this current work.

In conclusion, many cortical asymmetries seen later in life are already present at birth, and are minimally associated with preterm birth. Moreover, associations between biological sex and cortical asymmetries only appear to emerge after the neonatal period. Marked differences in cortical thickness asymmetries between neonates and young adults may represent developmental milestones that occur through experience-dependent mechanisms such as intracortical myelination. However, as our results are based on two cross-sectional cohorts, longitudinal studies across the lifespan, beginning in the perinatal period, are required to fully address these hypotheses.

## Methods

### Participants

**Developing Human Connectome Project:** Neonates were participants in the dHCP, approved by the National Research Ethics Committee (REC: 14/Lo/1169), and were scanned at the Evelina Newborn Imaging Centre, Evelina London Children's Hospital between 2015 and 2019. Written consent was obtained from all participating families prior to imaging. Term neonates were included if they were born between $37^{+0}$ and $41^{+6}$ weeks GA, if they were scanned $37^{+0}$ weeks PMA, and if both structural and functional data were available. Preterm neonates were included if they were born $<37^{+0}$ weeks GA, if they were scanned $37^{+0}$ weeks PMA, and if both structural and functional data were available. Both term and preterm neonates were excluded if their scans had incidental findings with possible/likely significance for both clinical and imaging analysis (e.g. destructive white matter lesions). However, incidental findings with possible clinical significance but unlikely analysis significance were permissible (125).

**Human Connectome Project:** Our choice of adult dataset was motivated by quality (current state-of-the-art in healthy, young adults) and size (a sample size that is still

considered large, and is amenable to vertex-wise permutation testing at sufficiently high resolution, which we define as 32,492 vertices per hemisphere). We included 1110 healthy young adults (605 biological females) who participated in the HCP-YA, approved by the internal review board of Washington University in St. Louis (IRB #201204036). These participants were recruited from ~300 families of twins and their non-twin siblings (126), provided written informed consent prior to imaging and were scanned at Washington University in St. Louis between 2012 and 2015. Inclusion and exclusion criteria for the HCP-YA can be found in (126) (Supplementary Table 1) - of note, preterm birth was one of these exclusion criteria. No statistical methods were used to predetermine sample sizes in either cohort.

## MRI Acquisition

**Developing Human Connectome Project:** MRIs were acquired in a single scan session for each neonate using a 3-Tesla Philips Achieva system (Philips Medical Systems, Best, The Netherlands). Full details regarding the preparation of neonates for scanning have been previously described (127). In brief, all neonates were scanned without sedation in a scanner environment optimised for neonatal imaging, including a dedicated 32-channel neonatal coil. MR-compatible ear putty and earmuffs were used to provide additional noise attenuation. Neonates were fed, swaddled and positioned in a vacuum jacket prior to scanning to promote natural sleep. All scans were supervised by a neonatal nurse and/or paediatrician who monitored heart rate, oxygen saturation and temperature throughout the scan (127).

T2w scans were acquired with a repetition time/echo time (TR/TE) of 12s/156ms, SENSE=2.11/2.58 (axial/sagittal) with in-plane resolution of $0.8 \times 0.8$mm, slice thickness of 1.6mm and overlap of 0.8mm. Images were motion corrected and super-resolved to produce a final resolution of 0.5mm isotropic (full details in Cordero-Grande et al. (128)). fMRI scans were acquired over 15 minutes and 3 seconds (2300 volumes) using a multislice gradient-echo echo planar imaging sequence with multiband excitation (multiband factor 9). TR/TE was 392ms/38ms, flip angle was 34°, and the acquired spatial resolution was 2.15mm isotropic (129, 130).

**Human Connectome Project:** Full details regarding image acquisition have been previously reported (103, 131). Images were acquired on a customised 3-Tesla Siemens Skyra (Siemens AG, Erlangen, Germany) using a 32-channel head coil. T1w scans (3D MPRAGE) were acquired with a TR/TE of 2400 ms/2.14 ms, and inversion time of 1000 ms, and flip angle of 8°. T2w scans (Siemens SPACE) were acquired with a TR/TE of 3200 ms/565 ms (103).

## Surface Extraction and Cortical Feature Generation

**Developing Human Connectome Project:** The full details of the structural pipeline are described in Makropoulos et al. (124). In summary, motion-corrected, reconstructed T2w images were first bias-corrected and brain-extracted. Following this, images were segmented into different tissue types using the Draw-EM algorithm (124). Next, topologically correct white matter surfaces were fit first to the grey-white tissue boundary and then to the grey-

white interface of the MR intensities (132). Pial surfaces were generated by expanding each white matter mesh towards the grey-cerebrospinal fluid interface (124, 132). Separately, an inflated surface was generated from the white surface through a process of inflation and smoothing. This inflated surface was then projected to a sphere for surface registration. The structural pipeline generated a number of univariate features summarising cortical morphology. Of these we used: cortical thickness (estimated as the Euclidean distance between corresponding vertices on the white and pial surfaces); sulcal depth (estimates of mean surface convexity/concavity (124)); and surface area (calculated using the pial surface), estimated for each vertex as one third the area of each triangle it is a part of (133). Cortical thickness and surface area were corrected for folding bias by regressing out cortical curvature (134, 135), which is the same approach as the HCP minimal preprocessing pipeline (103).

**Human Connectome Project:** We used minimally preprocessed structural data, which is described in full by Glasser et al. (103). Briefly, the PreFreeSurfer pipeline involves: correcting MR gradient nonlinearity-induced distortions; brain extraction; image intensity bias correction using the method described in Glasser et al. (134); and correcting readout distortion. The distortion- and bias-corrected structural volumes were then passed through the Freesurfer pipeline which is a bespoke version of the FreeSurfer `-recon-all` command (136) that adjusts placement of the pial surface by exploiting intensity differences at the pial-cerebrospinal fluid boundary between T1w and T2w images. This modification ensures that lightly myelinated cortical grey matter is not artefactually excluded (103). The FreeSurfer pipeline generates measures of sulcal depth, cortical curvature and uncorrected cortical thickness. Surface area (calculated in the same manner as for the dHCP), and cortical thickness were corrected for folding bias as above.

### Creating Left-Right Symmetric Surface Atlas

A prerequisite for investigating cortical asymmetry at a vertex-wise level is registration to a single template space with left-right vertex-wise correspondence. Moreover, to account for rapid morphological changes of the neonatal cortex, registration to a template space should be achieved using template-to-template registration with a spatiotemporal atlas. To achieve this, the pre-existing dHCP spatiotemporal atlas (36) was extended, both to increase the age range covered by the template from 36-44 weeks PMA to 28-44 weeks PMA, and to enforce left-right symmetry of cortical shape (Figure 6). Figures S9 to S12 show population average templates for the white matter surfaces, sulcal depth, cortical thickness and T1w/T2w ratio maps spanning 28 to 44 weeks PMA. Templates were generated following initial rigid (rotational) alignment of all neonatal data to the HCP-YA fs_LR 32k template space. Preliminary neonatal templates were generated by averaging within each postmenstrual week using adaptive-kernel weighting (36). These were refined through repeated non-linear registration of examples to their closest weekly template using MSM (52, 53), until the variability of the templates converged. Bias towards the adult reference was removed by applying the inverse of the average affine transformations on the template and dedrifting the template (137, 138). Data used to generate these templates were neonates included as part of the 3rd dHCP release (139), which included 887 scan sessions from 783 neonates (578 healthy term-born; 683 singletons; 360 females). Neonates had a median GA at birth of 39.0

weeks (IQR: 34.4 - 40.5; range: 23.0 - 43.6), and median PMA at scan of 40.9 weeks (IQR: 38.6 - 42.3; range: 26.7 - 45.1). Full demographic data for this data release are available online (see Methods: Data Availability).

Left-right vertex correspondence was achieved using MSM (52, 53) to co-register the left and right sulcal depth maps from each postmenstrual week of the atlas. In each case, registration was run in left-to-right and right-to-left directions (replicating the approach used in Garcia et al. (51)). Deformation maps of the left-to-right and the inverse of the right-to-left registrations were then averaged to generate an intermediate symmetric template space onto which all data (sulcal depth, cortical thickness, T1w/T2w ratio maps, and anatomical meshes) for a given postmenstrual week were resampled. Despite vertex correspondence between the left and right hemispheres, residual asymmetries persist due to differences in sulcal morphology. These residual asymmetries were removed by averaging the left and right sulcal depth templates (Figure 6).

Finally, registration was performed between all consecutive symmetric templates, for example from 30 to 31 weeks and 31 to 32 weeks, in order to generate a set of spherical transformations that, once concatenated, allow direct mapping from each local PMA template to the 40-week PMA template (34, 35).A further resampling step was then performed to get local template metrics and anatomical meshes into 40-week PMA symmetric template space. As the fs_LR 32k template used in the HCP-YA pipeline already has vertex correspondence (140), residual asymmetries were removed by averaging the left and right sulcal depth templates.

## Registration to Symmetric Surface Atlas

**Developing Human Connectome Project:** To accurately map structural asymmetry across all parts of the cortex, registration was run from all subjects' native space to their local symmetric template space, using MSM (52, 53) optimised for alignment of sulcal depth features (see Methods: Code Availability). The subject native-to-local PMA template registration deformation was combined with the local-to-40 week PMA registration deformation to create a single native-to-40 week PMA registration deformation for each example. Subject metrics and anatomical meshes were subsequently resampled from their native space to the 40-week PMA template cortical surface in a single step using adaptive barycentric interpolation (103), using this native-to-40 week PMA registration deformation.

**Human Connectome Project:** Although structurally aligned cortical surface data (termed MSMSulc) have been publicly released as part of the HCP-YA, registration was optimised for functional alignment rather than folding alignment, and so offers weaker correspondence of cortical folds across subjects (138). Since the objective of this analysis was to assess the correspondence of structural asymmetries from birth to adulthood, native-to-template registration was re-run for the HCP-YA subjects using MSM (52, 53) optimised for correspondence of sulcal depth across subjects (see Methods: Code Availability). Subject metrics and anatomical meshes were resampled from their native space to the HCP-YA fs_LR 32k sulcal depth template using adaptive barycentric interpolation (103). Quality

control for all surface registrations was performed by visual inspection, and were deemed accurate.

**Neonatal to Adult Registration:** Comparison of structural cortical asymmetries between birth and adulthood was achieved by registering the symmetric dHCP 40-week PMA sulcal depth template to the symmetric HCP fs_LR 32k sulcal depth template using MSM (52, 53), creating a template-to-template registration deformation. This deformation was then used to resample single subject structural cortical asymmetry maps from neonatal to adult space. The quality of the dHCP-to-HCP template registration was visually inspected and deemed to be as accurate as individual subject-to-template registrations.

## rs-fMRI Preprocessing

We extended the dHCP functional pipeline (130) to the cortical surface by projecting individual subject rs-fMRI timeseries from the native rs-fMRI volume space to native subject surfaces warped into rs-fMRI volumetric space, using a ribbon-constrained volume-to-surface mapping method similar to the HCP pipeline (103). These surface rs-fMRI maps were then resampled onto the 40-week PMA symmetric surface template space using the MSM native-to-40 week PMA template space registration deformations above. After resampling to symmetric surface space, rs-fMRI timeseries were smoothed with a kernel of $\sigma = 4$mm.

Dimensionality reduction prior to group ICA was achieved using MELODIC's Incremental Group PCA (MIGP) (141) with a dimensionality set at 2000. Using a group ICA, dual-regression approach, asymmetry observed in a dual-regressed single subject spatial map does not represent asymmetry in that individual per se, but instead represents group asymmetry that is then projected onto that individual's data. To remove this bias from the dual-regressed, single subject spatial maps, we generated symmetric group components by concatenating the left hemispheric MIGP output to the right hemispheric MIGP output, and vice versa. Following the left-to-right and right-to-left concatenation of the MIGP output, surface-based group ICA was performed (set at a dimensionality of 25) (142). Group components were then dual-regressed (143, 144) to generate subject-level spatial maps. Of the 25 group components, 16 were labelled as signal (Figure S2). Of note, MIGP and group ICA were performed on the term neonatal cohort, and single subject spatial maps for the preterm cohort were generated by dual-regressing the term neonatal group components into the rs-fMRI timeseries of each preterm neonate scanned at TEA.

## Generating Asymmetry Maps

Asymmetry maps for each cortical imaging metric (structural and functional for dHCP, and structural only for HCP-YA) were generated by subtracting the right hemisphere metric from the corresponding left hemisphere metric, then normalising as $(L–R)/((L+R)/2))$. Structural asymmetry maps were smoothed with a kernel size of $\sigma = 2$mm to improve signal-to-noise ratio, but no further smoothing of the functional asymmetry maps was performed. Structural asymmetry maps were smoothed on subject-specific symmetric midthickness surfaces, which were created by flipping the right midthickness surface along the YZ plane, averaging coordinates of the left and right surfaces, and then smoothing the result using `wb_command-`

*Nat Hum Behav*. Author manuscript; available in PMC 2026 March 21.

`surface-smoothing` (145, 146) for 10 iterations with a strength of 0.75. This was to ensure that metric smoothing was not biased by either hemisphere. Individual functional asymmetry maps were then thresholded with masks that were generated by thresholding each symmetric group component at $Z > 5.1$, to ensure that all subject asymmetry maps for a given component had the same spatial extent, as required by FSL PALM (37, 147).

## Statistical Analysis

Statistical analysis was performed using surface-based, vertex-wise permutation testing (FSL PALM, version alpha119 (37, 147)) with threshold-free cluster enhancement (TFCE; (148)). GA at birth, PMA at scan, biological sex, birthweight Z-score, total brain volume and hemispheric volume asymmetry (left - right) were included as explanatory variables. As the time between birth and scan was variable, both GA and PMA were included as explanatory variables for the term asymmetry analysis. However, GA was removed in the preterm vs. term comparison, since the objective of this experiment was to test for associations between cortical asymmetry and GA at birth. As it cannot be assumed that all term neonates were healthy, birthweight Z-score was included alongside GA as an additional, albeit imperfect, explanatory variable to account for neonatal health at birth. Birthweight Z-scores were calculated using INTERGROWTH-21 growth curves (149), which returned birthweight Z-scores specific for GA and biological sex. Both total brain volume and hemispheric volume asymmetry were included as additional explanatory variables, as it has been shown that these are highly predictive of cortical asymmetries (91). When comparing cortical asymmetries between neonates and adults, biological sex, total brain volume and hemispheric volume asymmetry were included as explanatory variables.

TFCE was performed on the group average symmetric midthickness anatomical surface, with the default H (2.0) and E (0.6) parameters (148), which are also the same as those previously reported (19). These group midthickness surfaces were created by regenerating the subject-specific symmetric midthickness surfaces without smoothing, and then averaging across subjects. As surface area was important for calculating TFCE, the group average surface area of the midthickness surface was also included as an input for vertex-wise permutation testing. Six one-sample $t$-tests were used to investigate cortical asymmetry (including associations with biological sex and PMA) in the term neonates, (C1: left>right, C2: right > left, C3: increasing PMA, C4: decreasing PMA, C5: female > male, and C6: male > female), and two two-sample unpaired $t$-tests were used to investigate differences in asymmetry between term and preterm neonates at TEA (C1: term > preterm, and C2: preterm > term). Six one-sample $t$-tests were performed to investigate cortical structural asymmetry in adults (C1: left > right, C2: right > left, C3: female > male, and C4: male > female, C5: increasing age, and C6: decreasing age), and two two-sample unpaired $t$-tests were performed to investigate differences in asymmetries between neonates and adults (C1: neonates > adults, and C2: adults > neonates). Family-wise error rate corrections were applied to $-\log_{10}(p)$-values across image features and design contrasts. $-\log_{10}p$-values were computed based on 10,000 random shuffles, with statistical significance set at $-\log_{10}(p)_{mcfwe} > 1.3$ (equivalent to $p_{mcfwe} < 0.05$) (150, 151).

A one sample *t*-test was performed to assess hemispheric volume asymmetry in term and preterm dHCP cohorts separately, and the HCP-YA cohort. Two-sample *t*-tests were performed to compare continuous variables between preterm and term groups (GA, PMA, birthweight, birthweight Z-score, total brain volume, and hemispheric volume asymmetry), and a chi-squared independence test was performed to compare categorical variables (biological sex) between preterm and term cohorts. These analyses were performed using the Pingouin Python package (version 0.5.3) (152).

## Supplementary Material

Refer to Web version on PubMed Central for supplementary material.

## Acknowledgements

The authors would like to thank the participants and families recruited in the dHCP, and all the neonatal staff at the Evelina Newborn Imaging Center, St. Thomas' Hospital, Guy's & St. Thomas' NHS Foundation Trust, London, UK. The authors acknowledge use of the research computing facility at King's College London, Rosalind (https://rosalind.kcl.ac.uk), which is delivered in partnership with the National Institute for Health Research (NIHR) Biomedical Research Centres at South London & Maudsley and Guy's & St. Thomas' NHS Foundation Trusts, and part-funded by capital equipment grants from the Maudsley Charity (award 980) and Guy's & St. Thomas' Charity (TR130505). The views expressed are those of the authors and not necessarily those of the NHS, the NIHR, King's College London, or the Department of Health and Social Care.

The dHCP project was funded by the European Research Council (ERC) under the European Union Seventh Framework Programme [FR/2007–2013]/ERC Grant Agreement no. 319,456. This study was supported in part by the Wellcome Engineering and Physical Sciences Research Council Centre for Medical Engineering at King's College London [grant WT 203,148/Z/16/Z] and the Medical Research Council (UK) [grant MR/K006355/1].

Data were provided [in part] by the Human Connectome Project, WU-Minn Consortium (Principal Investigators: David Van Essen and Kamil Ugurbil; 1U54MH091657) funded by the 16 NIH Institutes and Centers that support the NIH Blueprint for Neuroscience Research; and by the McDonnell Center for Systems Neuroscience at Washington University.

L.Z.J.W is supported by funding from the Commonwealth Scholarship Commission, United Kingdom. E.C.R is supported by an Academy of Medical Sciences/the British Heart Foundation/the Government Department of Business, Energy and Industrial Strategy/the Wellcome Trust Springboard Award [SBF003/1116] and E.C.R. and S.M.S. are supported by a Wellcome Collaborative Award [215573/Z/19/Z]. J.O. and A.D.E. received support from the Medical Research Council Centre for Neurodevelopmental Disorders, King's College London [grant MR/N026063/1]. J.O. is supported by a Sir Henry Dale Fellowship jointly funded by the Wellcome Trust and the Royal Society [grant 206,675/Z/17/Z]. The funders had no role in study design, data collection and analysis, decision to publish or preparation of the manuscript.

## Data Availability

The images used to produce the figures presented here are available as scenes through https://balsa.wustl.edu/study/2xrBN. UKB asymmetry summary measures reported in Sha et al. (48) are available at https://archive.mpi.nl/mpi/islandora/object/mpi:1839_24c1553d_3ee8_4879_8877_79ca19a0ac6a?asOfDateTime=2021-11-02T14:30:48.830Z. The following templates are publicly available: dhcpSym spatiotemporal cortical surface atlas: https://brain-development.org/brain-atlases/atlases-from-the-dhcp-project/cortical-surface-template/; HCP sulcal depth template: https://github.com/Washington-University/HCPpipelines/tree/master/global/templates/standard_mesh_alases; deformations between HCP fs_LR and FreeSurfer *fsaverage* space: https://github.com/Washington-University/HCPpipelines/tree/master/global/templates/standard_mesh_atlases/resample_fsaverage. Demographic data for

the dHCP are available at https://github.com/BioMedIA/dHCP-release-notes/blob/master/supplementary_files/combined.tsv.

## Code Availability

The code used to perform image processing and asymmetry analyses is available at https://github.com/metrics-lab/CorticalAsymmetry. This study also utilised the following software and code: surface atlas creation (without symmetrisation): https://github.com/jelenabozek/SurfaceAtlasConstruction; dHCP structural pipeline: https://github.com/BioMedIA/dhcp-structural-pipeline and dHCP functional pipeline: https://git.fmrib.ox.ac.uk/seanf/dhcp-neonatal-fmri-pipeline/-/tree/master; symmetrising resting-state timeseries: https://git.fmrib.ox.ac.uk/seanf/asymmetry-analysis; surface registration, metric and anatomical mesh resampling: https://github.com/ecr05/dHCP_template_alignment; MSM: https://github.com/ecr05/MSM_HOCR/releases; dHCP MSM configuration file: https://github.com/ecr05/dHCP_template _alignment/blob/master/configs/config_subject_to_40_week_template_3rd_release; HCP MSM configuration file optimised for sulcal depth: https://github.com/metrics-lab/CorticalAsymmetry/blob/main/config_standard_MSMstrain_HCP_CorticalAsymmetry; FSL PALM: https://github.com/andersonwinkler/PALM; INTERGROWTH-21 growth curves: http://intergrowth21.ndog.ox.ac.uk/; Connectome Workbench: https://www.humanconnectome.org/software/connectome-workbench; Pingouin Python package: https://pingouin-stats.org/.

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

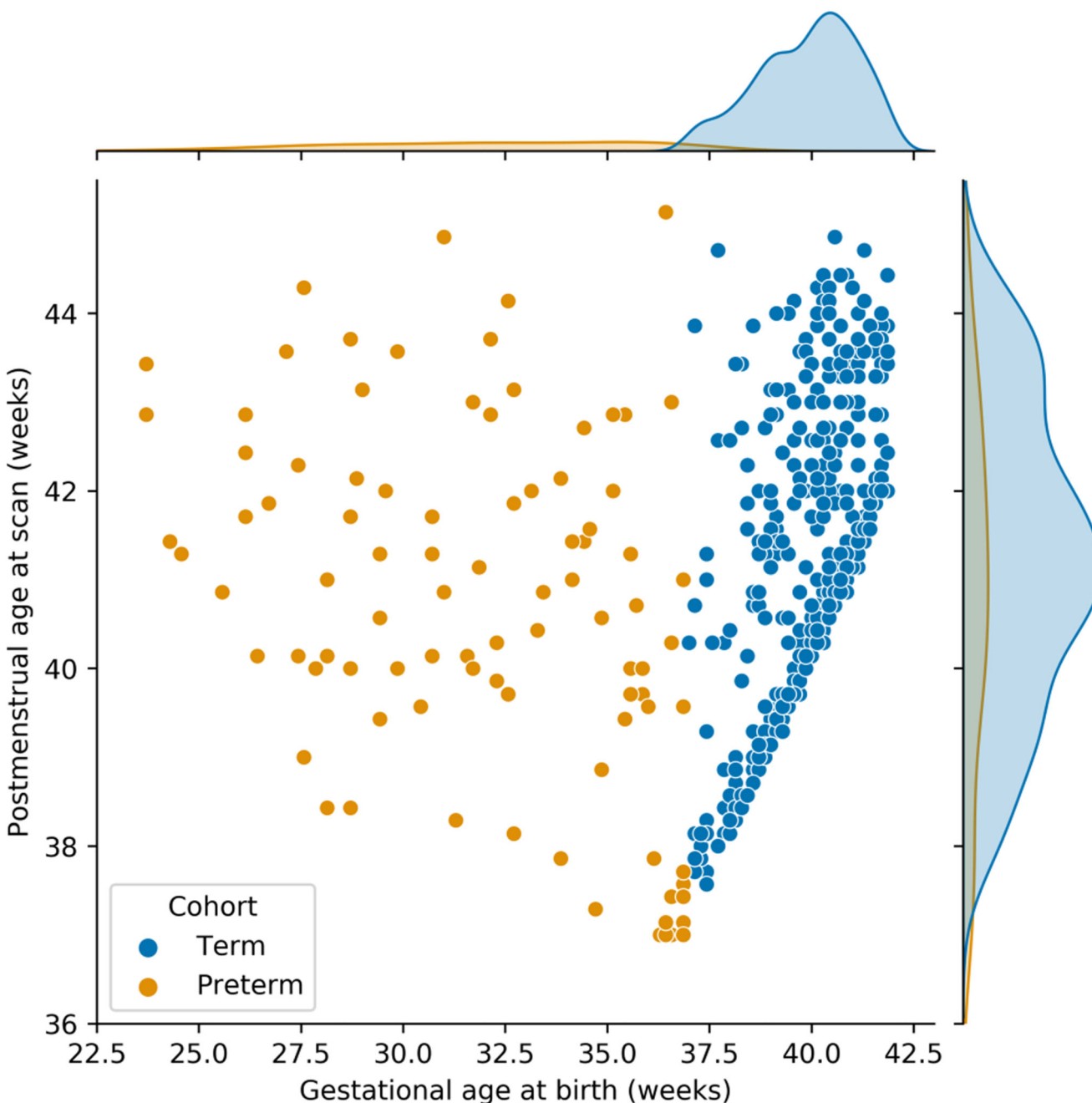

**Fig. 1. Gestational age at birth and postmenstrual age at scan for preterm and term-born neonates**

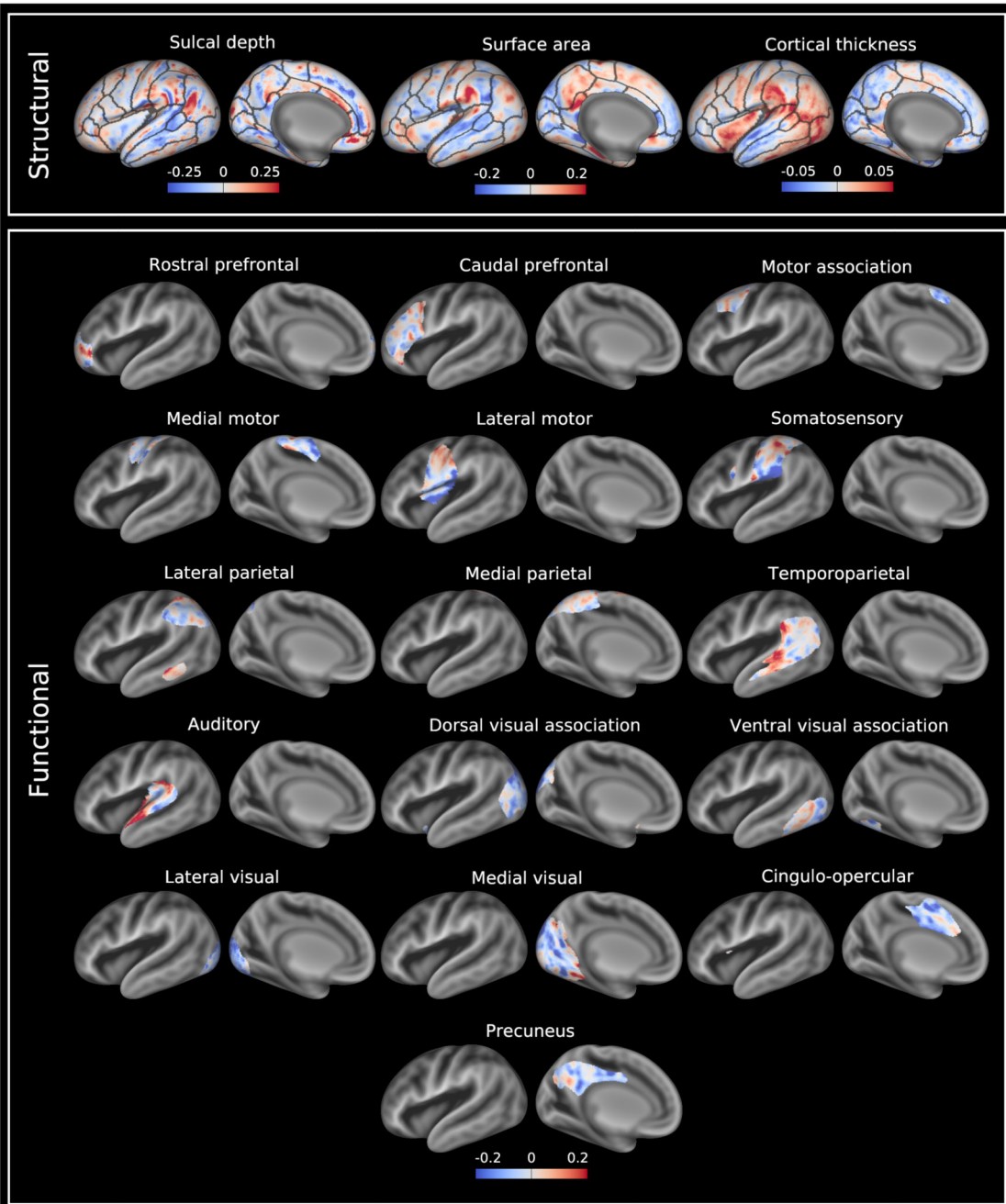

**Fig. 2. Median asymmetry indices of structural and functional asymmetries across the healthy term-born neonatal cohort scanned at term-equivalent age.**
Leftward asymmetries are color-coded red, and rightward asymmetries color-coded blue. Asymmetry indices are visualised on a very inflated 40-week PMA left hemispheric surface, and are overlaid on a 40-week PMA sulcal depth template (grey scale colour scheme). Anatomical regions of interest from a neonatal version of the Desikan-Killiany atlas (153, 154) are overlaid on the structural asymmetries for reference. Data available at https://balsa.wustl.edu/r2kN2.

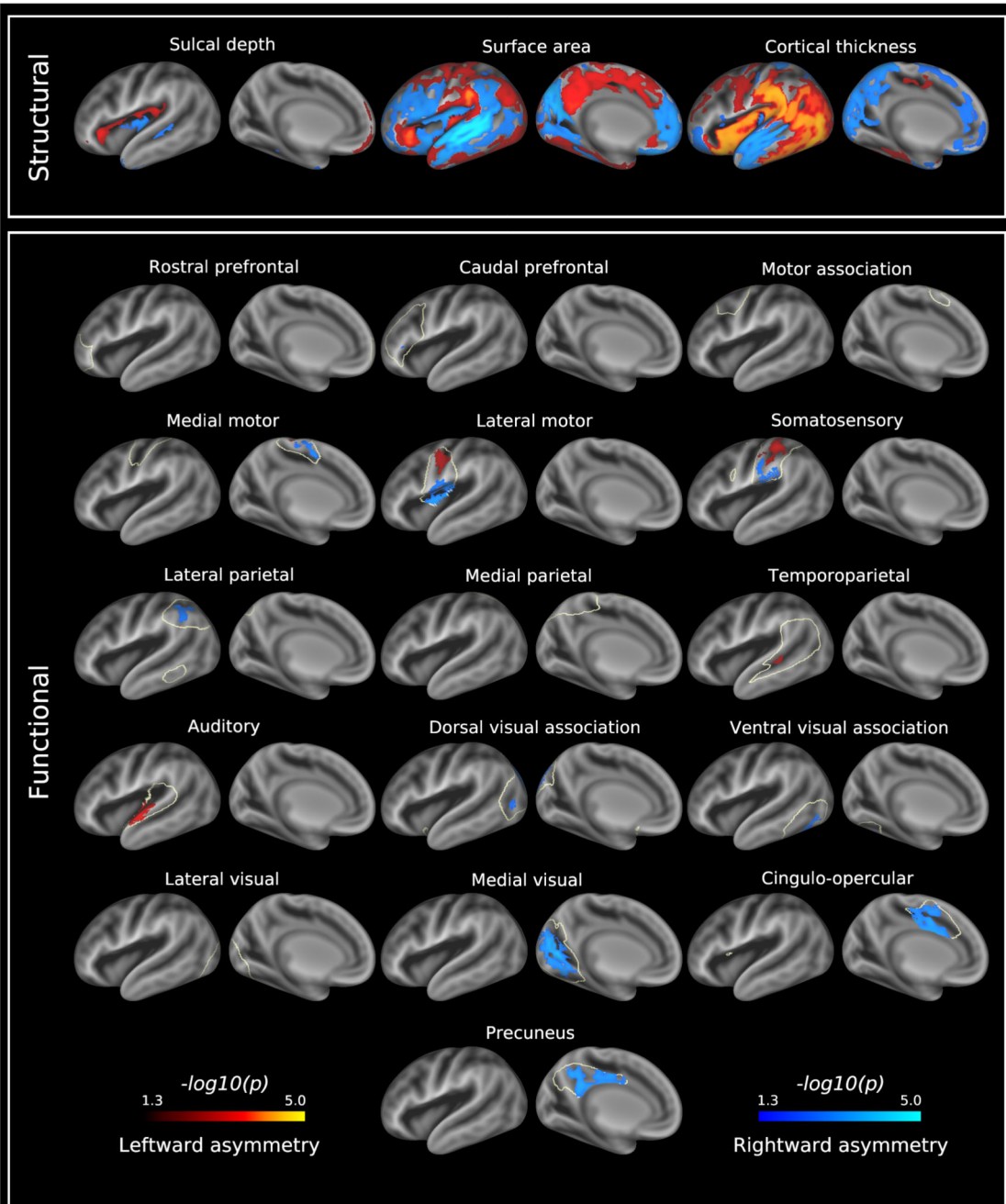

**Fig. 3. $-\log_{10}(p)_{mcfwe}$ value maps for structural and functional asymmetries in the healthy term-born neonatal cortex at term-equivalent age.**

Leftward asymmetries are represented by the red-yellow colour scale and rightward asymmetries by the blue-light blue colour scale. Significantly asymmetric regions are visualised on a very inflated 40-week PMA left hemispheric surface, and are overlaid on a 40-week PMA sulcal depth template (grey scale colour scheme). Off-white lines surrounding the functional asymmetries represent the mask used to threshold single subject asymmetry maps (see Methods: Generating Asymmetry Maps). Data at https://balsa.wustl.edu/x8509.

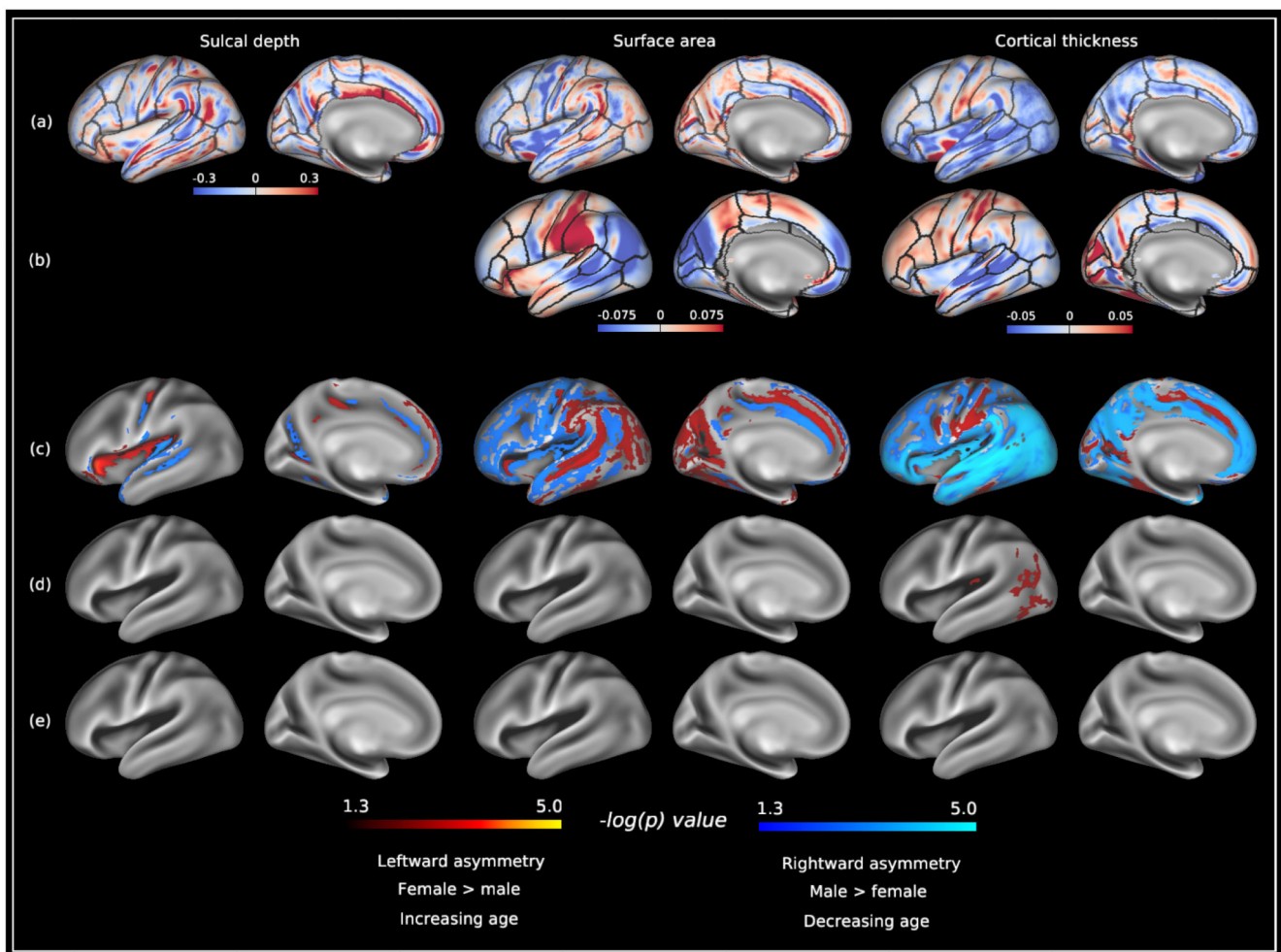

**Fig. 4. Structural asymmetries in the Human Connectome Project - Young Adult.**
(a) Median asymmetry indices of structural cortical measures across the HCP-YA cohort. (b) Mean asymmetry indices calculated in 31,864 UKB subjects made freely available by Sha et al. (48). Leftward asymmetries are color-coded red, and rightward asymmetries color-coded blue. Anatomical regions of interest from the Desikan-Killiany atlas (155) are overlaid on the structural asymmetries for reference. (c) $-\log_{10}(p)_{mcfwe}$ value maps for structural asymmetries in the HCP-YA. (d) $-\log_{10}(p)_{mcfwe}$ value maps for the effect of biological sex on cortical asymmetries in the HCP-YA. (e) $-\log_{10}(p)_{mcfwe}$ value maps for the effect of age on cortical asymmetries in the HCP-YA. Red-yellow color scale represents: (c) leftward asymmetries, (d) asymmetries in female > male, (e) and asymmetries linearly increasing with age. Blue-light blue color scale represents: (c) rightward asymmetries, (d) asymmetries in male > female, and (e) asymmetries decreasing linearly with age. Significantly asymmetric regions are visualised on a very inflated HCP-YA left hemispheric surface, and are overlaid on the HCP fs_LR sulcal depth template (grey scale colour scheme). Data at https://balsa.wustl.edu/P2DXn.

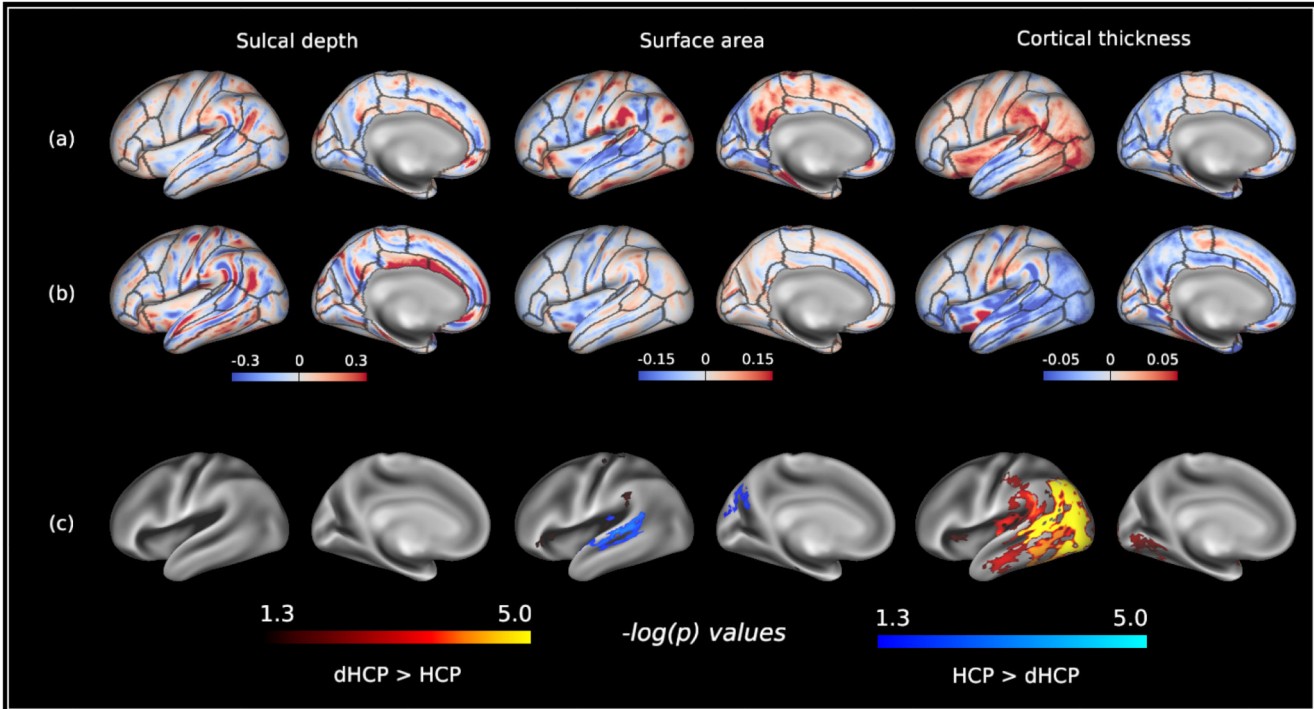

**Fig. 5. Comparison of structural asymmetries between the Developing Human Connectome Project and Human Connectome Project (Young Adult).**

(a) Median asymmetry indices of structural cortical measures across the dHCP cohort. (b) Median asymmetry indices of structural cortical measures across the HCP-YA cohort. (c) $-\log_{10}(p)_{mcfwe}$ value maps for differences in structural asymmetries between the dHCP and HCP-YA. Leftward asymmetry indices are color-coded red, and rightward asymmetry indices color-coded blue. Anatomical regions of interest from the Desikan-Killiany atlas (155) are overlaid on the structural asymmetries for reference. Asymmetry indices for the dHCP were resampled from dhcpSym space to HCP-YA fs_LR 32k space using template-to-template registration described in Methods: Registration to Symmetric Surface Atlas. dHCP > HCP-YA asymmetries are represented by the red-yellow colour scale and HCP-YA > dHCP asymmetries are represented by the blue-light blue colour scale. Data at https://balsa.wustl.edu/7xDlX.

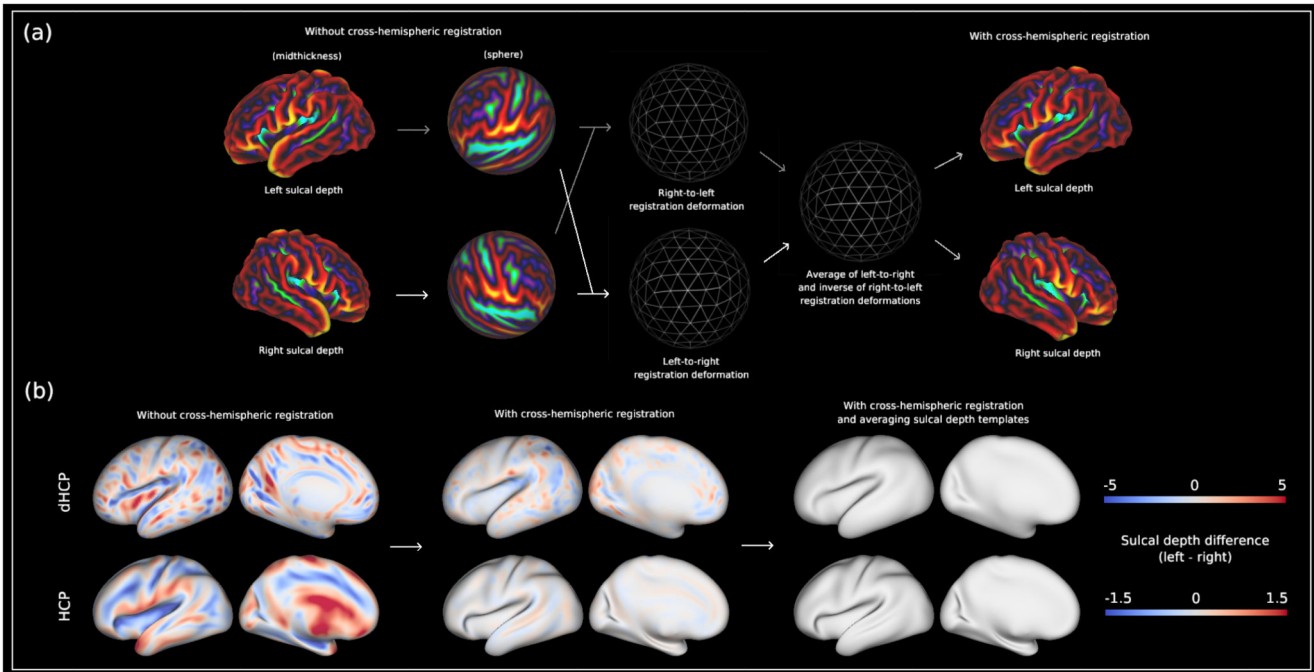

**Fig. 6.**

(a) Development of symmetric surface-based atlas. (i) Using MSM, the left and right hemispheres from each week of the non-symmetric spatiotemporal dHCP atlas were co-registered in order to generate left-right vertex correspondence of coarse scale patterns of cortical folding (sulcal depth). Registration was run in left-to-right and right-to-left directions. (ii) Deformation maps of the left-to-right and the inverse of the right-to-left registrations were averaged to generate an intermediate symmetric template space. (iii) Local non-symmetric template metrics and anatomical meshes were resampled into their corresponding symmetric template (dhcpSym). Consecutive registrations between local symmetric templates were then performed to achieve a local to 40-week PMA registration deformation. A further resampling step was performed to get local template metrics and anatomical meshes into 40-week PMA symmetric template space (dhcpSym40). (b) Difference between left and right sulcal depth maps without cross-hemispheric registration, with cross-hemispheric registration, and with cross-hemispheric registration + averaging left and right sulcal depth templates. For the HCP, the difference map without cross-hemispheric registration represents the difference between the left and right sulcal depth templates calculated in FreeSurfer *fsaverage* space, whereas the difference map with cross-hemispheric registration represents the difference between the left and right sulcal depth templates calculated in HCP fs_LR 32k space. Data at https://balsa.wustl.edu/64DBl.

**Table 1 Neonatal Demographics. Data presented as number (percentage), or median (interquartile range).**

| | Term (n = 442) | Preterm (n = 103) | *p*-value |
|---|---|---|---|
| Biological female | 200 (45.1) | 48 (46.6) | 0.99 |
| Gestational age at birth (weeks) | 40.1 (39.0 - 40.8) | 32.3 (29.2 - 35.3) | < 0.001 |
| Postmenstrual age at scan (weeks) | 41.1 (40.0 - 42.6) | 41.0 (39.6 - 42.2) | 0.075 |
| Birthweight (kg) | 3.37 (3.02 - 3.74) | 1.64 (1.15 - 2.39) | < 0.001 |
| Birthweight Z-score | 0.17 (−0.60 - 0.90) | −0.10 (−1.06 - 0.49) | < 0.001 |
| Total brain volume ($cm^3$) | 338.3 (308.6 - 370.5) | 340.8 (302.1 - 378.1) | 0.96 |
| Hemispheric volume asymmetry (left - right; $cm^3$) | −1.8 (−0.9 - −2.8) | −1.9 (−0.9 - −3.0) | 0.19 |

