## [Peer Review File · Nature human behaviour]

Peer Review Information

Journal: Nature Human Behaviour

Manuscript Title: Structural and functional asymmetry of the neonatal cerebral cortex

Corresponding author name(s): Logan Z. J. Williams and Emma C. Robinson

Reviewer Comments & Decisions:

Decision Letter, initial version:

16th December 2021

Dear Dr Williams,

Thank you once again for your manuscript, entitled "Structural and functional asymmetry of the neonatal cerebral cortex", and for your patience during the peer review process.

Your Article has now been evaluated by 4 referees. You will see from their comments copied below that, although they find your work of considerable interest, they have raised various concerns. In light of these comments, we cannot accept the manuscript for publication in the current form, but would be interested in considering a revised version if you are willing and able to fully address reviewer and editorial concerns.

We hope you will find the referees' comments useful as you decide how to proceed. If you wish to submit a substantially revised manuscript, please bear in mind that we will be reluctant to approach the referees again in the absence of major revisions. We are committed to providing a fair and constructive peer-review process. Do not hesitate to contact us if there are specific requests from the reviewers that you believe are technically impossible or unlikely to yield a meaningful outcome.

In particular, it will be important to carefully address the concern about the asymmetry of the atlas, raised by Ref #2 and #3, with further analysis e.g. following the suggestions of Ref #3.

Also, a number of the reviewers comment that the Results section is brief, and ask for a more extensive reporting of the results. We agree with the reviewers on this point and ask you to add the additional results requested by each of the reviewers. Please note that Nature Human Behaviour papers are allowed 8 display items (Figures + Tables) in the main text.

You should also carry out additional analysis to enable the quantitative comparison of neonatal structural and functional asymmetry with older age groups, e.g. by making use of existing public MRI datasets, as suggested by Ref #3 and others.

If you wish to submit a suitably revised manuscript we would hope to receive it within 6 months. We understand that the COVID-19 pandemic is causing significant disruptions which may prevent you from carrying out the additional work required for resubmission of your manuscript within this timeframe. If you are unable to submit your revised manuscript within 6 months, please let us know. We will be happy to extend the submission date to enable you to complete your work on the revision.

- Include a "Response to the editors and reviewers" document detailing, point-by-point, how you addressed each editor and referee comment. If no action was taken to address a point, you must provide a compelling argument. This response will be used by the editors to evaluate your revision and sent back to the reviewers along with the revised manuscript.
- Highlight all changes made to your manuscript or provide us with a version that tracks changes.

[REDACTED]

Thank you for the opportunity to review your work. Please do not hesitate to contact me if you have any questions or would like to discuss the required revisions further.

Sincerely,
Jamie

Dr Jamie Horder
Senior Editor
Nature Human Behaviour

REVIEWER COMMENTS:

Reviewer #1:
Remarks to the Author:

This is an excellent addition to the literature characterizing neonatal brain asymmetry, which goes beyond previous studies in various respects – including its relatively large sample size, analysis of both

structure and function, advanced methods of asymmetry measurement deriving from surface-based registration, and an age-appropriate spatiotemporal atlas. One important result is that brain asymmetries showed no significant differences in neonates born preterm. I have only a small number of queries/suggestions on the text:

From the Introduction: 'Previous work has investigated structural (15–22) and functional (23–25) asymmetries in the perinatal period and early infancy. As is seen in adults (20), term and preterm neonatal brains exhibit leftward volume (left greater than right) and sulcal depth (left deeper than right) asymmetry (19, 20, 22) in regions important for processing language'.

This does not seem to capture that the opposite asymmetry, i.e. right>left, has been reported for superior temporal sulcal depth (reference 21). The introductory summary of the literature may need to reflect inconsistencies (or perhaps this is just a writing error here, mixing left and right - or am I confused?). In fact, the present study finds a rightward asymmetry of superior temporal sulcal depth as well.

From the Introduction: 'By comparing cortical asymmetries between the same 442 term born neonates and 103 preterm neonates at term-equivalent age (TEA), we directly tested the impact of early life environmental perturbations on the development of structural and functional asymmetry in the newborn brain.'

I think that testing associations between preterm birth and brain asymmetries does not amount to directly testing the impact of early life environmental perturbations. It is true that preterm birth may arise from, or lead to, early life environmental perturbations that affect brain development. Then again, it may not in some/many cases. For example, altered brain asymmetry may sometimes arise from underlying genetic or random developmental processes, which could additionally contribute to the risk of preterm birth. I suggest to keep the issues distinct in the write-up, i.e., avoid confusing what was actually tested versus these kinds of cause/effect/etiology hypotheses and interpretations. In general, the manuscript would benefit throughout from a more cautious use of cause-effect language (e.g. the heading 'Effect of preterm birth' is loaded with a cause-effect hypothesis that might not be justified).

From the Discussion: 'Recent work investigating the genomic loci underpinning brain asymmetries has implicated genes that are expressed during prenatal brain development (53), which are involved in molecular pathways responsible for left-right axis patterning in vertebrates (53), visual and auditory pathway development (54), neural activity during development of the somatosensory cortex (55), and regulation of axonal guidance and synaptogenesis (56).'

I was able to identify genes from reference 53 that were studied in references 54 and 55, but could not obviously see a link to reference 56. Can the authors please clarify the connection?

Reviewer #2:

Remarks to the Author:

The authors can be congratulated to this very large sample of newborn data that indeed represent the largest dataset on neonatal brain imaging, acquired on 3T scanner and with a unique imaging protocol (using a super-resolution approach for structural imaging that involves acquiring overlapping slices of 1.6mm to achieve 0.8.mm isotropic resolution with reasonable scan time . this approach though not

providing true 0.8 mm out of plane resolution. fMRI is based on a larger image resolution of 2.15 mm isotropic.

The analysis presented here are based on a symmetric atlas described and published earlier but is said to be based on preterm infants with GA between 24-45 wks. I have not found the information in this paper as to how many fullterm infants generated the atlas. The main idea here was to project structural and resting-state data onto a template surface (atlas) and then perform all analysis and comparisons at the surface space, which certainly has some advantages over the traditional 3 D volumetric approach, but still induces a general bias of any atlas approach as surface morphology is not identical from patient to patient and which also lead to the not completely symmetric feature of the atlas described in limitations of the study. This remains a main concern as the initial template cannot be perfectly symmetric and this will influence the asymmetric measures and differences. The authors say they used the residuals for post-hoc testing but Fig. S7 and S8 I did not find them in the supplementary material! Maybe the mean S3-S4?

The paper clearly adds to the literature by combining asymmetry measurements both from structural features (sulcal depth, surface area, cortical thickness) and from resting state functional networks. I would though recommend the authors to refrain from stressing the "firstness" as many of the asymmetries the paper finds have been described many times before, actually there do not seem to be structural novel previously unrecognized asymmetries? In line 167 they state that they report unique results but then in line 177 they say the results are in line with previous studies. Not sure what is the real advantage by doing things in so complex ways.

What is new is the characterization of the asymmetries in the rs-fMRI and they seem to be both left and rightward in almost all rs-fMRI assessed. Even though the authors present interesting interpretations of this finding f.e. in the auditory network one wonders why this would be the case in all networks? to what extent the asymmetries are purely driven by the data and not by bias due to asymmetries in the template?

The rightward asymmetry found in the superior temporal region, which has been described in many newborns studies is found here too and one wonders if this asymmetry, which is in contrast to a language leftward asymmetry found later in life, is more related to a human specific rightward sulcal depth asymmetry described by Leroy et al PNAS 2015? It might be interesting to discuss this study in relation to the discussion of the genetic basis of asymmetries and the fact that preterm infants at term show these same rightward asymmetries, corresponding with the earlier appearance of the right superior temporal sulcus in development and the larger STS on the right than the left described in preterm infants by Dubois J et al 2008. This earlier study also described differences in female and male newborns, did the authors find any gender differences in their measures?

Overall it might be useful to make a bit clearer what this rather complicated analysis provided beyond and above the already described asymmetries in early structural and functional brain development. Have the authors also tried a subject space based assessment of asymmetries?

Reviewer #3:

Remarks to the Author:

The key result of this paper is an assessment of hemispheric asymmetries in neonates. Many asymmetries were found at term-equivalent age and they were not modulated by preterm birth. This manuscript stands to become an authoritative report with regards to structural and functional asymmetries in neonates.

The Developing Human Connectome Project dataset is by some margin the largest obtained. The pre-processing strategies appear valid: state-of-the-art analysis methods were used, to generate symmetric, age-specific surface-based brain templates, to which the brains of individual participants were then registered. One limitation the authors note (lines 326-330) is that there remain residual asymmetries in the template. Why could these not be eliminated entirely by mirroring and averaging the coordinates in the final template? I believe this will be unlikely to consequentially alter the results, but it would seem more elegant to eliminate this bias entirely.

The statistical methods appear appropriate and the results are robust.

A limitation of the project is that it does not attempt to compare the asymmetries in any formal way to those found in adults. As most sulci are present by TEA, and at almost 90% of their adult depth, reasonable registration between and adult template space (e.g., from the HCP) and the infant space is possible. This would allow a quantitative comparison of infant and adult asymmetries, for example by reporting them for each of the 360 Glasser et al (2016) parcellations, or, with the data from the manuscript's reference [46]. This would allow insight into which of the adult asymmetries are already present, and which are not, which is one of the most interesting questions.

A further limitation is that beyond the renderings (which are shared online), no region-by-region assessment of asymmetry is provided. Some readers may be unfamiliar, for example, with the exact location of primary auditory cortex, or which specific secondary auditory regions have the leftwards asymmetry. Registration with a standard adult template, and reporting of region-by-region values using a standard atlas, would improve the accessibility and reuse of the results.

Another aspect of the manuscript that could be further refined is the connection of the asymmetries to function. Where do the functional labels provided for the resting-state networks on Fig. 1 come from (e.g., "motor association")? Are these the authors interpretations? Or was there some quantitative assessment, for example by comparison with neurosynth.org? And, in the text some of the discussion seems a little loose. For example, on line 245 it is claimed that the unexpected rightward asymmetry of auditory cortex might be due to pitch processing. But pitch processing is quite symmetric (<https://neurosynth.org/analyses/terms/pitch/>) and more subtle differences are thought to differentiate processing in the hemispheres (e.g., <https://www.nature.com/articles/s41562-019-0548-z>).

Reviewer #4:

Remarks to the Author:

This paper studies structural and resting state functional MRI asymmetry of the cortex in a large and novel sample of 442 healthy newborn full-term infants (24-45 weeks old) as well as 103 pre-term

infants, all from one site of the Developing Human Connectome Project (DHCP). The bulk of the findings are shown in Figure 1 (the only actual results figure) that shows patterns of left/right asymmetry that are more widespread for structure (such as cortical thickness) versus various resting state networks. A secondary finding (left to a couple of supplemental figures) is the lack of difference in asymmetry in pre-term versus full-term infants at term equivalent age. Overall the paper is easy to read and highlights that cortical asymmetry is set early on in brain development.

Comments

1. There were 200/442 females in the full-term and 48/103 females in the pre-term. There is no sex analysis in the paper which seems unacceptable.
2. No asymmetry values, the main focus of the paper, are given in the core manuscript. How asymmetric are the structural and functional metrics and how do these values compare to studies in older children/adults?
3. As the paper is focused on asymmetry in the developing brain, it would seem that an asymmetry versus age analysis would be appropriate given they have a decent age span in both the full-term and pre-term cohorts – that would make a nice results Figure 2. As is, one does not get any impression of how asymmetry changes with age in the neonates.
4. Similar to the above comment, it is difficult to pull out how the whole brain asymmetry patterns actually compare to similar studies in older children, adolescents, young adults and older. For a paper with 114 citations, it seems surprising that they don't cite large sample papers that have actually examined cortical thickness asymmetry of whole brain across the lifespan from childhood to old age: e.g. Zhou D et al Neuroimage 2013 – Cortical thickness asymmetry from childhood to older adulthood - n=274, cross-sectional, 5-59 years; Li G et al J Neuroscience 2015 – Spatial patterns, longitudinal development and hemispheric asymmetries of cortical thickness in infants from birth to 2 years of age – n=73, longitudinal 0-2 years (Note: this paper is Ref 19 but is referred to in the Discussion for only one brain region, the middle superior temporal sulcus for some reason – what about the rest of the brain?); Plessen K et al J Neuroscience 2014 – Sex, age, and cognitive correlates of asymmetries in thickness of the cortical mantle across the lifespan – n=215, 7-59 years; etc. It would be useful to have more direct discussions on left/right asymmetry seen in the neonates with this and other similar studies. The Discussion, paragraph 2 seems to be the only paragraph on these MRI study comparisons, but it seems rather limited with a main focus on middle superior temporal sulcus and very little on other brain regions, only one reference by Sha ref 46 (now in PNAS).
5. The conclusion seems to be stepping outside the bounds of the results shown in the paper. Would it not be better to focus on the relationship of asymmetries between structure and function in neonates and how this all compares to older healthy participant asymmetries in cortical structure and resting state function?

Author Rebuttal to Initial comments

We are grateful for the reviewer comments and suggestions. We have now made substantial changes to the manuscript in line with the recommendations. These include:

- Further symmetrisation of the neonatal surface template by averaging the left and right sulcal depth templates for each postmenstrual week
- Investigating associations between cortical asymmetries, postmenstrual age and biological sex
- Comparison of structural cortical asymmetries between healthy-term neonates and healthy young adults from the Human Connectome Project, using template-to-template registration

In addition, we have made changes to the statistical analyses that improve the robustness of the results presented in our manuscript. These include:

- Adjusting for total brain volume and hemispheric volume asymmetry, as recommended by Williams, C. M., et al. (2022). Comparing brain asymmetries independently of brain size, *NeuroImage*
- Smoothing structural cortical measures on individual surfaces instead of a global template
- Increasing the Z-statistic threshold used to define the functional ROI masks required by FSL PALM, to ensure the functional asymmetries observed were 'real' rather than the result of a liberal functional ROI mask

Reviewer #1:

Remarks to the Author:

This is an excellent addition to the literature characterizing neonatal brain asymmetry, which goes beyond previous studies in various respects – including its relatively large sample size, analysis of both structure and function, advanced methods of asymmetry measurement deriving from surface-based registration, and an age-appropriate spatiotemporal atlas. One important result is that brain asymmetries showed no significant differences in neonates born preterm. I have only a small number of queries/suggestions on the text:

From the Introduction: *'Previous work has investigated structural (15–22) and functional (23–25) asymmetries in the perinatal period and early infancy. As is seen in adults (20), term and preterm neonatal brains exhibit leftward volume (left greater than right) and sulcal depth (left deeper than right) asymmetry (19, 20, 22) in regions important for processing language.'*

1. **Comment:** This does not seem to capture that the opposite asymmetry, i.e. right>left, has been reported for superior temporal sulcal depth (reference 21). The introductory summary of the literature may need to reflect inconsistencies (or perhaps this is just a writing error here, mixing left and right - or am I confused?). In fact, the present study finds a rightward asymmetry of superior temporal sulcal depth as well.

Response: A similar comment (comment #19) made by reviewer #4. The authors agree that the overview of the previous literature was too brief, which came at the expense of its accuracy. We have therefore included a more detailed summarisation of the literature concerning cortical asymmetries in the neonatal period.

- *Introduction (Lines 33 - 56):* For example, one of the most well-characterised structural asymmetries of the human cerebral cortex is the right-deeper-than-left asymmetry of the superior temporal sulcus, which has been demonstrated in fetuses (15, 16), preterm (17, 18) and term neonates (19, 20), children and adults (19, 21, 22). Other sulcal depth asymmetries that have been described include the rightward asymmetry of the parieto-occipital sulcus in neonates (20), and adults (19), and the leftward asymmetry of the lateral fissure (18–20).

In contrast, asymmetries of surface area and cortical thickness during the neonatal period have received less attention. Li et al. demonstrated leftward surface area asymmetries in the precentral, postcentral, middle frontal and cingulate gyri, temporal pole and insula; as well as rightward surface area asymmetry in the inferior parietal, posterior temporal and lateral occipital cortices (20); and rightward cortical thickness asymmetries in the temporal and lateral occipital cortices, and insula (23). Functional asymmetries, measured using task-based functional MRI (fMRI), are also present at birth and during infancy, and include lateralised responses to speech (24, 25) and music (26).

From the Introduction: *‘By comparing cortical asymmetries between the same 442 term born neonates and 103 preterm neonates at term-equivalent age (TEA), we directly tested the impact of early life environmental perturbations on the development of structural and functional asymmetry in the newborn brain.’*

2. **Comment:** I think that testing associations between preterm birth and brain asymmetries does not amount to directly testing the impact of early life environmental perturbations. It is true that preterm birth may arise from, or lead to, early life environmental perturbations that affect brain development. Then again, it may not in some/many cases. For example, altered brain asymmetry may sometimes arise from underlying genetic or random developmental processes, which could additionally contribute to the risk of preterm birth. I suggest to keep the issues distinct in the write-up, i.e., avoid confusing what was actually tested versus these kinds of cause/effect/etiology hypotheses and interpretations. In general, the manuscript would benefit throughout from a more cautious use of cause-effect language (e.g. the heading ‘Effect of preterm birth’ is loaded with a cause-effect hypothesis that might not be justified).

Response: The authors agree that more thoughtful wording was required, and have updated the text to reflect our findings more accurately. Specifically, we refrain from suggesting ‘effects’ of preterm birth, postmenstrual age and biological sex on cortical asymmetry. Instead we report associations (or lack thereof) between cortical asymmetries and these demographic variables. The following changes have been made referencing preterm birth:

- *Abstract (Lines 14-15):* while associations with preterm birth
- *Results (Line 197):* Heading is now changed from “Effects of Preterm Birth” to “Preterm Birth”
- *Results (Lines 204-207):* we directly tested for associations between preterm birth and the development of structural and functional asymmetry in the newborn cerebral cortex.
- *Conclusion (Lines 597-598):* and are minimally associated with preterm birth.

From the Discussion: *‘Recent work investigating the genomic loci underpinning brain asymmetries has implicated genes that are expressed during prenatal brain development (53), which are involved in molecular pathways responsible for left-right axis patterning in vertebrates (53), visual and auditory pathway development (54), neural activity during development of the somatosensory cortex (55), and regulation of axonal guidance and synaptogenesis (56).’*

3. **Comment:** I was able to identify genes from reference 53 that were studied in references 54 and 55, but could not obviously see a link to reference 56. Can the authors please clarify the connection?

Response: Thank you for the comment. We refer the reviewer to the following excerpt from Sha et al. (2020). The genetic architecture of left-right structural brain asymmetry. *Nature Human Behaviour*.

"On 20p12.1, rs6135555 is in a region having a chromatin interaction with the *FLRT3* promoter in neural progenitor cells (Supplementary Fig. 4), a gene which regulates axon guidance and excitatory synapse development."

Reviewer #2:

Remarks to the Author:

The authors can be congratulated to this very large sample of newborn data that indeed represent the largest dataset on neonatal brain imaging, acquired on 3T scanner and with a unique imaging protocol (using a super-resolution approach for structural imaging that involves acquiring overlapping slices of 1.6mm to achieve 0.8mm isotropic resolution with reasonable scan time . this approach though not providing true 0.8 mm out of plane resolution. fMRI is based on a larger image resolution of 2.15 mm isotropic.

4. **Comment:** The analysis presented here are based on a symmetric atlas described and published earlier but is said to be based on preterm infants with GA between 24-45 wks. I have not found the information in this paper as to how many fullterm infants generated the atlas.

Response: Thank you for your comment. We would like to clarify that this new template was generated using data from the third data release of the Developing Human Connectome Project, of which a majority of participants are healthy term-born neonates. The authors agree that it was important to include a demographic summary of the neonatal subjects that contributed to the template, and have also included details on how the templates for each postmenstrual week were generated in the methods:

- **Methods (Lines 738 - 759):** Figures S9 to S12 show population average templates for the white matter surfaces, sulcal depth, cortical thickness and T1w/T2w ratio maps spanning 28 to 44 weeks PMA. Templates were generated following initial rigid (rotational) alignment of all neonatal data to the HCP-YA fs_LR 32k template space. Preliminary neonatal templates were generated by averaging within each postmenstrual week using adaptive-kernel weighting (36). These were refined through repeated non-linear registration of examples to their closest weekly template using MSM (52, 53), until the variability of the templates converged. Bias towards the adult reference was removed by applying the inverse of the average affine transformations on the template and dedrifting the template (130, 131). Data used to generate these templates were neonates included as part of the 3rd dHCP release (132), which included 887 scan sessions from 783 neonates (578 healthy term-born; 683 singletons; 360 females). Neonates had a median GA at birth of 39.0 weeks (IQR: 34.4 - 40.5; range: 23.0 - 43.6), and median PMA at scan of 40.9 weeks (IQR: 38.6 - 42.3; range: 26.7 - 45.1). Full demographic data for this data release are available online (see Methods: Data Availability).
5. **Comment:** The main idea here was to project structural and resting-state data onto a template surface (atlas) and then perform all analysis and comparisons at the surface space, which certainly has some advantages over the traditional 3 D volumetric approach, but still induces a general bias of any atlas approach as surface morphology is not identical from patient to patient and which also lead to the not completely symmetric feature of the atlas described in limitations of the study. This remains a main concern as the initial template cannot be perfectly symmetric and this will influence the asymmetric measures and differences. The authors say they used the residuals for post-hoc testing but Fig. S7 and S8 I did not find them in the supplementary material! Maybe the mean S3-S4?

Response: We thank the reviewers for their comments. Residual asymmetry in the template was a concern shared by reviewer #2 and #3. We therefore adopted the suggestion of reviewer #3 (comment 11), who proposed to remove any residual folding asymmetry in the template by averaging the sulcal depth template of the left and right hemispheres. Figure 6 has now been updated to show the impact of this additional symmetrising step for both the dHCP and HCP surface templates (additional experiments with older age groups were suggested by reviewers 3 and 4). We also resampled the sulcal depth map of the fs_LR template back to fsaverage space to highlight the differences in sulcal depth asymmetry before and after vertex correspondence in the HCP. As the template space for all analyses no longer have residual asymmetries, we have removed post-hoc analyses assessing the impact of these residual asymmetries. The following text has been added:

- *Methods (Lines 770 - 774):* Despite vertex correspondence between the left and right hemispheres, residual asymmetries persist due to differences in sulcal morphology. These residual asymmetries were removed by averaging the left and right sulcal depth templates (Figure 6).
 - *Methods (Lines 782 - 785):* As the fs_LR 32k template used in the HCP-YA pipeline already has vertex correspondence (133), residual asymmetries were removed by averaging the left and right sulcal depth templates.
6. **Comment:** The paper clearly adds to the literature by combining asymmetry measurements both from structural features (sulcal depth, surface area, cortical thickness) and from resting state functional networks. I would though recommend the authors to refrain from stressing the “firstness” as many of the asymmetries the paper finds have been described many times before, actually there do not seem to be structural novel previously unrecognized asymmetries? In line 167 the state that they report unique results but then in line 177 they say the results are in line with previous studies. Not sure what is the real advantage by doing things in so complex ways.

Response: The authors agree that the paper overemphasised the firstness of the results. We also acknowledge that the description and discussion of the structural asymmetries was limited and therefore not clear. Both the results and discussion have been rewritten to highlight the new structural asymmetries described, without overemphasising firstness.

- *Results (Lines 113 - 141):* Figure 2 and Figure 3 show asymmetry indices and $-\log_{10}(p)$ mcfwe value maps, respectively, for sulcal depth, surface area and cortical thickness. Only a limited number of sulci demonstrated significant asymmetry ($-\log_{10}(p)$ mcfwe >1.3) after registration to a fully symmetric template (Methods: Creating Left-Right Symmetric Surface Atlas). Leftward asymmetry (left deeper than right) was observed for the anterior and posterior lateral fissure. There was also rightward asymmetry (right deeper than left) of the middle portion of the superior temporal sulcus. Regarding surface area, there were leftward asymmetries (left > right) located in the supramarginal, medial precentral, postcentral, posterior cingulate, caudal anterior cingulate, caudal superior frontal and inferior temporal gyri, superior lateral occipital lobe and anterior insula. There were rightward asymmetries (right > left) located in the superior and middle temporal, lingual, rostral anterior cingulate, rostral superior frontal, middle frontal, and inferior frontal gyri, insula and parieto-occipital sulcus.

Compared to the asymmetry indices for sulcal depth and surface area, cortical thickness asymmetry indices were smaller in magnitude (Figure 2). There were leftward cortical thickness asymmetries (left > right) located in the supramarginal, angular, precentral, postcentral, superior frontal, and inferior temporal gyri, lateral occipital lobe and insula. There were also rightward asymmetries (right > left) located in the superior temporal, middle temporal, caudal superior frontal and anterior cingulate gyri, orbitofrontal cortex,

precuneus, and calcarine and parieto-occipital sulci.

- *Discussion (Lines 306 - 339)*: Our results not only replicate findings reported in earlier neonatal studies, but also identify new asymmetries that have previously been observed in older cohorts. The rightward depth asymmetry of the superior temporal sulcus and leftward depth asymmetry of the lateral fissure presented here are in agreement with previous studies in preterm neonates both before and at TEA (18). Of the surface area asymmetries reported by Li et al., we replicate leftward asymmetry of the postcentral and caudal cingulate gyri, temporal pole and anterior insula, and rightward asymmetry in the inferior parietal, posterior temporal and inferior lateral occipital cortices (20). We also replicate cortical thickness asymmetries observed in the superior and middle temporal gyri (23).

Our results further build upon earlier studies by identifying novel asymmetries in both surface area and cortical thickness. Regarding surface area, we observed leftward asymmetry in the superior lateral occipital lobe, and rightward asymmetry spanning the superior frontal, middle frontal, inferior frontal and rostral anterior cingulate gyri, medial occipital lobe, and mid-insula. For cortical thickness, we found leftward asymmetry of the inferior temporal, supramarginal, angular, precentral and postcentral gyri, lateral occipital lobe and insula, and rightward asymmetry of the caudal superior frontal and anterior cingulate gyri, orbitofrontal cortex, precuneus, and calcarine and parieto-occipital sulci. Some of these newly recognised asymmetries have been described in older cohorts. For instance, Zhou et al. demonstrated rightward asymmetry of cortical thickness in the inferior frontal gyrus and medial occipital lobe, and leftward asymmetry of cortical thickness in the supramarginal and angular gyri, and lateral occipital lobe in children and adolescents (46). More recently, Roe et al. demonstrated rightward surface area asymmetry of the medial occipital lobe and caudal anterior cingulate gyrus from as early as 4 years of age (47).

7. **Comment:** What is new is the characterization of the asymmetries in the rs-fMRI and they seem to be both left and rightward in almost all rs-fMRI assessed. Even though the authors present interesting interpretations of this finding f.e. I the auditory network one wonders why this would be the case in all networks ? to what extent the asymmetries are purely driven by the data and not by bias due to asymmetries in the template ?

Response: We thank the reviewer for their comment. After addressing the residual sulcal depth asymmetries in the template, the auditory network no longer demonstrated both left and rightward asymmetries. The results and discussion has been updated to reflect this.

- *Results (Lines 156 - 158)*: Asymmetry in the auditory network comprised a leftward asymmetric region located along the anterior end of the superior temporal gyrus.
- *Discussion (Lines 340 - 351)*: Although functional asymmetries have not previously been investigated using surface-based analyses, our findings are consistent with volumetric-based studies in neonates and adults. Studies investigating language using speech stimuli in task-based fMRI have shown leftward asymmetric activation along the peri-Sylvian region in neonates (24, 25), children (54) and adults (2, 3). The leftward asymmetry of the auditory network seen here is also consistent with leftward asymmetries in auditory networks observed in children (54) and adults (55), and might be related to a leftward asymmetric increase in fractional anisotropy of the arcuate fasciculus that has been observed in neonates (29, 56) and adults (57, 58).

However, somatosensory and motor networks still display this pattern of organisation, which we still believe is related to somatotopic organisation of the sensorimotor networks. We have built upon our discussion of the leftward asymmetry seen in the upper limb homuncular representation:

- *Results (Lines 170 - 180)*: In the medial motor network, a leftward asymmetric region was located on the dorsal aspect of the central sulcus, and a rightward asymmetric region on the dorsomedial aspect of the central sulcus. The lateral motor network contained a leftward asymmetric region located in the midportion of the central sulcus and a rightward asymmetric region located on the inferolateral aspect of the central sulcus and dorsal mid-insula. For the somatosensory network, the leftward asymmetric region was located in the mid-portion of the postcentral gyrus, and the rightward asymmetric region on the inferolateral aspect of the postcentral gyrus.
- *Discussion (Lines 392 - 412)*: In addition to recapitulating well-documented functional asymmetries in the auditory and visual systems seen in adulthood, we observed a pattern of functional asymmetry in the somatomotor networks that might reflect the somatotopic organisation of the sensorimotor cortex (69–71). Namely, the somatosensory and lateral motor networks showed leftward asymmetry in the upper limb region of the somatotopic map (69). This functional asymmetry might be associated with microstructural asymmetry (increased fractional anisotropy) of the left corticospinal tract compared to the right, which has previously been demonstrated in neonates (29), adolescents (72) and adults (73). It has also been shown that leftward lateralisation of motor network connectivity was associated with better motor performance in children (74), and that motor circuit connectivity in children diagnosed with autism spectrum disorder was more rightward asymmetric compared to children without autism spectrum disorder (75). This asymmetry may also relate to handedness, as genomic loci associated with handedness (76, 77) are also associated with brain asymmetry and are expressed during prenatal brain development (78).

We also discuss the rightward asymmetry seen in the somatomotor networks:

- *Discussion (Lines 413 - 427)*: The somatosensory and lateral motor networks also demonstrated rightward asymmetry in the mouth region in the somatomotor homunculus which, in neonates, includes the mid-dorsal insula (69). We hypothesise that this rightward asymmetry is important for the development of swallowing in the neonatal period. Swallowing is a highly complex motor function that is represented across a number of cortical areas including the primary somatomotor cortices (79) and mid-insula (80, 81). Moreover, swallowing is initiated in response to a number of stimuli (79), two of which (taste and interoception), are represented in the right mid-dorsal insula (82). Together, this rightward asymmetry seen in neonates could reflect integration of sensorimotor, taste and interoceptive information that is critical for the development of swallowing.
8. **Comment:** The rightward asymmetry found in the superior temporal region, which has been described in many newborns studies is found here too and one wonders if this asymmetry, which is in contrast to a language leftward asymmetry found later in life, is more related to a human specific rightward sulcal depth asymmetry described by Leroy et al PNAS 2015? It might be interesting to discuss this study in relation to the discussion of the genetic basis of asymmetries and the fact that preterm infants at term show these same rightward asymmetries, corresponding with the earlier appearance of the right superior temporal sulcus in development and the larger STS on the right than the left described in preterm infants by Dubois J et al 2008.

Response: The authors agree that further discussion on the rightward STS asymmetry was warranted. We have updated the discussion surrounding the STS asymmetry, linking to the functional asymmetries seen in the auditory and temporoparietal networks, and the potential mechanisms driving this asymmetry:

- *Discussion (Lines 352 - 371)*: Additionally, we identified a region of leftward asymmetry in the temporoparietal network centred on the middle portion of the superior temporal sulcus. Together, the asymmetries of the auditory and temporoparietal networks share the same anatomical landmarks as the anterior and middle temporal voice areas, respectively (59). These areas constitute part of a functional network responsible for processing voice (59, 60) that appears to be closely associated with the rightward depth asymmetry of the superior temporal sulcus (59, 61). The presence of this rightward superior temporal sulcus asymmetry in humans across the lifespan (in both health and disease), but not in non-human primates, suggests that this asymmetry might be related to the evolution of human-specific cognitive abilities such as language (21). Moreover, the depth asymmetry of the superior temporal sulcus has been associated with two genes (DACT1 and DAAM1), which are mainly expressed during prenatal brain development (62). These findings lend further support to the importance of cortical asymmetries in the peri-Sylvian region for language development in early life.
9. **Comment:** This earlier study also described differences in female and male newborns, did the authors find any gender differences in their measures?

Response: The absence of any investigation in the effects of biological sex was a concern raised by reviewer #4 (comment #16). After re-running analyses to address this, we did not find any effect of biological sex on structural or functional asymmetry in neonates (Figures S4 and S5).

- *Results (Lines 191-195)*: Figure S4 shows the difference in asymmetry indices between biological females and males (female - male) for all cortical measures. No statistically significant associations between biological sex and structural or functional cortical asymmetries were observed (Figure S5).

We also updated the introduction to include background on previous literature in neonates investigating the effect of biological sex on cortical asymmetries:

- *Introduction (Lines 57 - 66)*: Although biological sex has been associated with numerous measures of neonatal brain structure and function (27), studies investigating associations between cortical asymmetries and biological sex are limited. Dubois et al. found an association between biological sex and the emergence of sulci, with sulci on the right hemisphere emerging significantly earlier than the left in biological females, but not males (17). However, Li et al. found no association between biological sex and cortical thickness asymmetries in a longitudinal cohort of neonates scanned at birth, 1 and 2 years of age (23).

These findings are also discussed:

- *Discussion (Lines 443 - 455)*: In contrast to earlier work (17, 20), we observed no associations between biological sex and structural or functional asymmetries in the neonatal period. Recent work by Williams et al. demonstrated that both total brain volume and hemispheric volume asymmetry were important explanatory variables when investigating differences in cortical asymmetries between biological sexes (91). The absence of these covariates in previous analyses may be one explanation of why the results presented here differ. Our results also suggest that observed sex-related differences in cortical asymmetry (1, 21, 91) likely emerge in later life. Further research is required to address when and how these sex-related differences in asymmetries emerge.

- Comment:** Overall it might be useful to make a bit clearer what this rather complicated analysis provided beyond and above the already described asymmetries in early structural and functional brain development. Have the authors also tried a subject space based assessment of asymmetries?

Response: Thank you for the comment. Our interpretation of subject space-based asymmetries is that asymmetries are calculated at a region-of-interest level, since vertex-wise comparisons are not possible in native space. This suggestion is similar to that made by reviewer #3 (comment #14). Although ROIs are useful for improving the accessibility of results, vertex-wise asymmetries rarely follow standard anatomical ROIs and have prompted the most recent asymmetry analyses to consider primarily vertex-wise assessment. So, we do not formally assess or report asymmetry indices by ROIs. More biologically meaningful ROIs such as the HCP_MMP1.0 template would be very useful in assessing cortical asymmetries in neonates (suggested by reviewer #3, comment #13). However, as outlined in our response to reviewer #3 (comment #13), propagating the HCP_MMP1.0 labels to neonates assumes that the features used to generate these labels in adults are identical in neonates, which is not the case.

- Discussion (Lines 583 - 600):* An appealing alternative was to drive registration in a multimodal manner, and assess both structural and functional asymmetries using the HCP_MMP1.0 atlas (116). However, as the intersubject correspondence of secondary and tertiary sulci is relatively poor following multimodal alignment (32, 116), such an approach would have limited the ability to detect sulcal depth asymmetries compared to the approach used here (Figure S8). Moreover, multimodal registration of neonatal cortical surfaces to the HCP_MMP1.0 template assumes that registration may be driven with the same cortical features (resting state networks, T1w/T2w ratio and visuotopic maps) as used for adults, which requires that they are present and have comparable topography in neonates. However, current evidence suggests that this is not the case (117, 118). Ultimately, multimodal registration of neonates to the adult HCP_MMP1.0 template is non-trivial, and outside the scope of this current work.

Reviewer #3:

Remarks to the Author:

The key result of this paper is an assessment of hemispheric asymmetries in neonates. Many asymmetries were found at term-equivalent age and they were not modulated by preterm birth. This manuscript stands to become an authoritative report with regards to structural and functional asymmetries in neonates.

The Developing Human Connectome Project dataset is by some margin the largest obtained. The pre-processing strategies appear valid: state-of-the-art analysis methods were used, to generate symmetric, age-specific surface-based brain templates, to which the brains of individual participants were then registered.

- Comment:** One limitation the authors note (lines 326-330) is that there remain residual asymmetries in the template. Why could these not be eliminated entirely by mirroring and averaging the coordinates in the final template? I believe this will be unlikely to consequentially alter the results, but it would seem more elegant to eliminate this bias entirely.

Response: Thank you for your comment and very helpful suggestion. This has been addressed in response to reviewer #2, comment #5. To briefly summarise, the left and right sulcal depth maps were averaged (after left-right vertex correspondence was achieved) to eliminate residual asymmetries.

- Comment:** The statistical methods appear appropriate and the results are robust.

Response: The authors would like to point out an adjustment to the statistical analyses that we believe improves the robustness of the results. A recent publication by CM Williams et al. (2022), *NeuroImage*, titled “Comparing brain asymmetries independently of brain size” systematically assessed the impact of additional covariates on investigating the effect of biological sex on cortical asymmetries. These authors found that including total brain volume and hemispheric volume asymmetry significantly influenced results. As such, we revised our methods to include these covariates:

- *Methods (Lines 881 - 883):* GA at birth, PMA at scan, biological sex, birthweight Z-score, total brain volume and hemispheric volume asymmetry (left - right) were included as explanatory variables.
- *Methods (Lines 895 - 901):* Both total brain volume and hemispheric volume asymmetry were included as additional explanatory variables, as it has been shown that these are highly predictive of cortical asymmetries (91). When comparing cortical asymmetries between neonates and adults, biological sex, total brain volume and hemispheric volume asymmetry were included as explanatory variables.

Additionally, an image processing decision that we believe influences the analyses is the surface smoothing step described on lines 861 - 864. Previously this surface smoothing was performed in template space, which is suboptimal. In the present manuscript, all structural asymmetry maps were smoothed on an individual surface:

- *Methods (Lines 864 - 871):* Structural asymmetry maps were smoothed on subject-specific symmetric midthickness surfaces, which were created by flipping the right midthickness surface along the YZ plane, averaging coordinates of the left and right surfaces, and then smoothing the result using `wb_command -surface-smoothing` (139, 140) for 10 iterations with a strength of 0.75. This was to ensure that metric smoothing was not biased by either hemisphere.

We also increased the Z-statistic threshold used to define the functional ROI masks required by FSL PALM, to be sure the functional asymmetries observed were ‘real’ rather than a result of a liberal functional ROI mask. The threshold for the functional ROI mask was increased from $Z > 3.1$ to $Z > 5.1$:

- *Methods (Lines 875 - 880):* Individual functional asymmetry maps were then thresholded with masks that were generated by thresholding each symmetric group component at $Z > 5.1$, to ensure that all subject asymmetry maps for a given component had the same spatial extent, as required by FSL PALM (37, 141).

13. **Comment:** A limitation of the project is that it does not attempt to compare the asymmetries in any formal way to those found in adults. As most sulci are present by TEA, and at almost 90% of their adult depth, reasonable registration between and adult template space (e.g., from the HCP) and the infant space is possible. This would allow a quantitative comparison of infant and adult asymmetries, for example by reporting them for each of the 360 Glasser et al (2016) parcellations, or, with the data from the manuscript’s reference [46]. This would allow insight into which of the adult asymmetries are already present, and which are not, which is one of the most interesting questions.

Response: We thank the reviewer for their comment and suggestions. Similar issues were raised by reviewer #4 (comment 20). We addressed these by comparing structural asymmetries between Developing Human Connectome Project (dHCP) and Human Connectome Project (Young Adult), using template to template registration:

- *Methods (Lines 816 - 823)*: Comparison of structural cortical asymmetries between birth and adulthood was achieved by registering the symmetric dHCP 40-week PMA sulcal depth template to the symmetric HCP fs_LR 32k sulcal depth template using MSM (52, 53), creating a template-to-template registration deformation. This deformation was then used to resample single subject structural cortical asymmetry maps from neonatal to adult space.

Structurally aligned HCP cortical data are publicly available (termed 'MSMSulc'), however, this registration is only an initialisation for the multimodal alignment used by Glasser et al. (2016), and is not appropriate for investigating structural asymmetries. We therefore re-ran cortical surface registration based on folding, before any analysis of structural asymmetry within the HCP-YA and between dHCP and HCP:

- *Methods (Lines 802 - 815)*: Although structurally aligned cortical surface data (termed MSMSulc) have been publicly released as part of the HCP-YA, registration was optimised for functional alignment rather than folding alignment, and so offers weaker correspondence of cortical folds across subjects (132). Since the objective of this analysis was to assess the correspondence of structural asymmetries from birth to adulthood, native-to-template registration was re-run for the HCP-YA subjects using MSM (52, 53) optimised for correspondence of sulcal depth across subjects (see Methods: Code Availability). Subject metrics and anatomical meshes were resampled from their native space to the HCP-YA fs_LR 32k sulcal depth template using adaptive barycentric interpolation (103).

Following this step we statistically compared structural asymmetry between the dHCP and HCP-YA population, and separately included summary measures of structural asymmetries from the UK Biobank population (calculated from Sha et al. (2021), PNAS). The results section has been updated accordingly:

- *Results (Lines 231 - 284)*: Finally, despite the number of studies assessing the trajectory of cortical asymmetries across the lifespan (45–47), there is little knowledge of how the cortical asymmetries established at birth differ from those seen in later life. To address this, we compared structural cortical asymmetries seen in the term neonatal cohort with those in 1110 healthy young adults from the Human Connectome Project - Young Adult (HCP-YA). Participants had a median (interquartile range (IQR)) age at scan of 29 (26 to 32) years, and total brain volume of 999.0 cm³ (925.5 to 108.0), and median hemispheric volume asymmetry of -7.9cm³ (-5.0 to -10.6, $p < 0.001$).

Figure 4a demonstrates the asymmetry indices of structural asymmetries in the HCP-YA cohort. Significant sulcal depth asymmetries were observed for the lateral fissure (leftward), and superior temporal and parieto-occipital sulci (rightward). Leftward surface area asymmetries were observed in the lateral postcentral, inferior temporal and superior frontal gyri, lateral and medial occipital lobe, and superior temporal sulcus. Rightward surface area asymmetries were observed in the superior temporal, middle temporal and cingulate gyri, frontal lobe and insula. Cortical thickness asymmetry was largely rightward asymmetric, with several regions of leftward asymmetry localised to the precentral, postcentral, inferior temporal and parahippocampal gyri, and cingulate and calcarine sulci (Figure 4c). Region-of-interest summaries of these structural metrics are provided in Figure S1. There were no linear associations with age (Figure 4e) but biological sex was significantly associated with cortical thickness asymmetry, with biological females having increased leftward asymmetry in the posterior middle and inferior temporal gyri, and lateral occipital lobe compared to biological males (Figure 4d).

Figure 4b also summarises previously published results from Sha et al. (see Methods: Data Availability), who performed vertex-wise cortical thickness and surface area asymmetry analysis for 31,864 data sets from the UK Biobank (UKB) (48–50). These asymmetry maps are smoother than those from our own analyses, as they are calculated on lower resolution surfaces (10,252 vertices per hemisphere, compared to 32,492 vertices per hemisphere used here) (48). Leftward surface area asymmetries were observed in the

postcentral, supramarginal and superior frontal gyri. Rightward surface area asymmetries were observed in the posterior temporal, lateral occipital and inferior parietal lobes. For cortical thickness, leftward asymmetries were observed for the postcentral gyrus, frontal lobe and cingulate sulcus. Rightward cortical thickness asymmetries were seen in the superior temporal, middle temporal and supramarginal gyri, lateral occipital and inferior parietal lobes, and parieto-occipital sulcus.

Finally, Figure 5 displays results comparing structural asymmetries between neonates and adults. The magnitude of asymmetry indices were similar between both populations (Figure 5a and Figure 5b). Young adults had higher surface area asymmetry indices (left > right) compared to neonates along the length of the superior temporal sulcus/gyrus, and superior portion of the parieto-occipital sulcus. In contrast, neonates had higher cortical thickness asymmetry indices (left > right) in the supramarginal and lingual gyri, lateral occipital, inferior parietal, and temporal cortices, and anterior and posterior insula. No differences in sulcal depth asymmetries were observed between these two groups (Figure 5c).

The results of within HCP asymmetry were discussed:

- *Discussion (Lines 496 - 524):* Assessing structural asymmetry within the HCP-YA was performed as an initial step towards investigating differences between neonates and adults. We also qualitatively compared these results against mean structural asymmetry maps calculated in 31,864 UKB participants (48). Although these average maps bear some resemblance to our own results, they also demonstrate important differences including 1) leftward surface area asymmetry in medial occipital lobe in HCP-YA, but rightward in UKB and 2) rightward asymmetry in frontal lobe in HCP-YA but leftward in UKB. This is perhaps unsurprising as a number of important differences exist between this study and ours in terms of image acquisition, processing and analysis. Specifically, the input data in this study is of higher resolution, which is important as Glasser et al. demonstrated the benefit of reducing voxel size from 1mm isotropic to 0.7mm isotropic on the accuracy of cortical thickness measurements, especially in areas such as the medial occipital lobe where cortical ribbon is thinner (103). We also performed statistical analysis on higher resolution surfaces (32,492 compared to 10,252 vertices), which likely explains the higher resolution features of the asymmetry analysis in Figure 4 relative to Sha et al. (48). Although Roe et al. reported high correlation of vertex-wise structural asymmetries between the HCP-YA and UKB on higher resolution cortical surfaces (47), surface extraction using only T1-weighted (T1w) images (rather than both T1w and T2-weighted (T2w) images) and high spatial smoothing were image processing decisions that did not fully capitalise on the quality of the HCP-YA data (103).

dHCP vs HCP asymmetry results discussed:

- *Discussion (Lines 525 - 552):* Unlike sulcal depth, the direction of surface area and cortical thickness asymmetries significantly differed between birth and adulthood. These asymmetry reversals might represent neurodevelopmental milestones. Shaw et al. demonstrated that the pattern of cortical thickness asymmetries observed in the orbitofrontal (left > right) and lateral occipital (right > left) cortices during childhood reversed by adulthood. Moreover, the reversal in orbitofrontal cortical thickness asymmetry between childhood and adulthood was disrupted in those with attention-deficit hyperactivity disorder (104). Previous longitudinal studies assessing cortical asymmetries (45–47, 104) have not considered the perinatal period, which is one of the most dynamic phases of brain development (105).

As our results are cross-sectional, longitudinal studies mapping cortical asymmetries from the perinatal period onwards will be important for characterising asymmetry reversals in typical development, and how they relate to the acquisition of new behaviours.

Such studies may also shed light on how these asymmetries emerge. For example, it has been suggested that cortical thickness asymmetries are, in part, experience-dependent and arise through processes such as intracortical myelination (47) (which is inversely correlated with cortical thickness (106)). On the other hand, surface area asymmetries are thought to reflect the organisation of cortical minicolumns established prenatally (47), with postnatal changes potentially occurring as a result of genetically-programmed developmental processes (107), such as apoptotic cell death (108).

We did not attempt any formal analysis of functional asymmetry between dHCP and HCP, nor did we assess dHCP vs. HCP asymmetries using the HCP MMP1.0 template. Given the poor correspondence between cortical folding and function, multimodal alignment would have precluded a thorough investigation of sulcal depth asymmetry. Moreover, registering neonatal cortical surfaces to the HCP_MMP1.0 template assumes that the same cortical features used to drive registration in the HCP-YA (40 RSNs, 1 T1w/T2w ratio map and 9 visuotopic maps) are identical in neonates, whereas current evidence suggests that they are not. Ultimately, while we believe assessing neonatal cortical asymmetries in the context of a multimodal atlas would be beneficial, it is beyond the scope of the current work. This is discussed in the text:

- *Discussion (Lines 578 - 600):* An important methodological consideration in this study was the modality driving surface registration. Cortical surfaces were registered to a symmetric template using sulcal depth maps, which may have impacted the investigation of functional asymmetry, particularly in cortical regions where structural and functional correspondence is poor (116). An appealing alternative was to drive registration in a multimodal manner, and assess both structural and functional asymmetries using the HCP_MMP1.0 atlas (116). However, as the intersubject correspondence of secondary and tertiary sulci is relatively poor following multimodal alignment (32, 116), such an approach would have limited the ability to detect sulcal depth asymmetries compared to the approach used here (Figure S8). Moreover, multimodal registration of neonatal cortical surfaces to the HCP_MMP1.0 template assumes that registration may be driven with the same cortical features (resting state networks, T1w/T2w ratio and visuotopic maps) as used for adults, which requires that they are present and have comparable topography in neonates. However, current evidence suggests that this is not the case (117, 118). Ultimately, multimodal registration of neonates to the adult HCP_MMP1.0 template is non-trivial, and outside the scope of this current work.

14. **Comment:** A further limitation is that beyond the renderings (which are shared online), no region-by-region assessment of asymmetry is provided. Some readers may be unfamiliar, for example, with the exact location of primary auditory cortex, or which specific secondary auditory regions have the leftwards asymmetry. Registration with a standard adult template, and reporting of region-by-region values using a standard atlas, would improve the accessibility and reuse of the results.

Response: Thank you for the comment. The authors agree that summarising asymmetry indices by ROIs would improve accessibility and have included an additional supplementary figure for this purpose (Figure S1). We have also included standard anatomical ROIs as outlines in all figures reporting asymmetry indices. However, these asymmetries rarely follow standard anatomical ROIs. So, we do not formally assess or report asymmetry indices by ROIs.

For neonates:

- *Results (Lines 141 - 143)*: As a reference, asymmetry indices for these structural measures are also reported on a per-region basis (Figure S1).

For adults:

- *Results (Lines 248 - 249)*: Region-of-interest summaries of these structural metrics are provided in Figure S1.

15. **Comment:** Another aspect of the manuscript that could be further refined is the connection of the asymmetries to function. Where do the functional labels provided for the resting-state networks on Fig. 1 come from (e.g., “motor association”)? Are these the authors interpretations? Or was there some quantitative assessment, for example by comparison with neurosynth.org?

Response: Thank you for the comment. The naming of RSNs observed in neonates is based on naming conventions of previous studies investigating RSNs in neonates e.g. Doria et al. (2010), PNAS; Fransson et al. (2007), PNAS, Eyre et al. (2021), Brain. RSNs not previously observed in neonates were named according to their anatomical location.

Comment: And, in the text some of the discussion seems a little loose. For example, on line 245 it is claimed that the unexpected rightward asymmetry of auditory cortex might be due to pitch processing. But pitch processing is quite symmetric (<https://neurosynth.org/analyses/terms/pitch/>) and more subtle differences are thought to differentiate processing in the hemispheres (e.g., <https://www.nature.com/articles/s41562-019-0548-z>).

Response: Thank you for the comment. We no longer observe leftward and rightward asymmetry of the auditory network. Regarding the other asymmetries observed here, we believe there is sufficient evidence to make connections between the location and direction of these asymmetries and potential functions. We have also searched Neurosynth, and believe that these results still hold. When looking at meta-analytic functional maps, we defined asymmetry as higher Z statistic in the cortical region of one hemisphere compared to the other, rather than spatial extent, as this is most similar to how asymmetry is defined in the current study. We include the following links for the reviewer:

- Activations associated with the terms "language" and "voice" appear leftward asymmetric (see <https://neurosynth.org/analyses/terms/language/> and <https://neurosynth.org/analyses/terms/voice/>).
- Activity in both the dorsal attention network, and MT+ are rightward asymmetric (see <https://neurosynth.org/analyses/terms/dorsal%20attention/> and <https://neurosynth.org/analyses/terms/visual%20motion/>).
- Activations associated with the term "face recognition" in Neurosynth also appear rightward asymmetric (<https://neurosynth.org/analyses/terms/face%20recognition/>).
- Although activations associated with the terms "hand" and "arm" in Neurosynth appear leftward asymmetric (<https://neurosynth.org/analyses/terms/hand/> and <https://neurosynth.org/analyses/terms/arm/>), this could likely reflect a bias in fMRI studies where samples are biased by the predominance of right-handedness within the population, rather than a true functional

asymmetry.

Reviewer #4:

Remarks to the Author:

This paper studies structural and resting state functional MRI asymmetry of the cortex in a large and novel sample of 442 healthy newborn full-term infants (24-45 weeks old) as well as 103 pre-term infants, all from one site of the Developing Human Connectome Project (DHCP). The bulk of the findings are shown in Figure 1 (the only actual results figure) that shows patterns of left/right asymmetry that are more widespread for structure (such as cortical thickness) versus various resting state networks. A secondary finding (left to a couple of supplemental figures) is the lack of difference in asymmetry in pre-term versus full-term infants at term equivalent age. Overall the paper is easy to read and highlights that cortical asymmetry is set early on in brain development.

16. **Comment:** There were 200/442 females in the full-term and 48/103 females in the pre-term. There is no sex analysis in the paper which seems unacceptable.

Response: This has been addressed in response to comment #9.

After re-running analyses to address this, we did not find any effect of biological sex on structural or functional asymmetry in neonates (Figures S4 and S5).

- *Results (Lines 191-195):* Figure S4 shows the difference in asymmetry indices between biological females and males (female - male) for all cortical measures. No statistically significant associations between biological sex and structural or functional cortical asymmetries were observed (Figure S5).

We also updated the introduction to include background on previous literature in neonates investigating the effect of biological sex on cortical asymmetries:

- *Introduction (Lines 57 - 66):* Although biological sex has been associated with numerous measures of neonatal brain structure and function (27), studies investigating associations between cortical asymmetries and biological sex are limited. Dubois et al. found an association between biological sex and the emergence of sulci, with sulci on the right hemisphere emerging significantly earlier than the left in biological females, but not males (17). However, Li et al. found no association between biological sex and cortical thickness asymmetries in a longitudinal cohort of neonates scanned at birth, 1 and 2 years of age (23).

These findings are also discussed:

- *Discussion (Lines 443 - 455):* In contrast to earlier work (17, 20), we observed no associations between biological sex and structural or functional asymmetries in the neonatal period. Recent work by Williams et al. demonstrated that both total brain volume and hemispheric volume asymmetry were important explanatory variables when investigating differences in cortical asymmetries between biological sexes (91). The absence of these covariates in previous analyses may be one explanation of why the results presented here differ. Our results also suggest that observed sex-related

differences in cortical asymmetry (1, 21, 91) likely emerge in later life. Further research is required to address when and how these sex-related differences in asymmetries emerge.

17. **Comment:** No asymmetry values, the main focus of the paper, are given in the core manuscript. How asymmetric are the structural and functional metrics and how do these values compare to studies in older children/adults?

Response: Thank you for this comment. We have now included figures of asymmetry indices in the term neonatal cohort (Figure 2), HCP-YA cohort (Figure 4) and term neonatal vs HCP-YA cohort (Figure 5).

Structural asymmetry indices:

- *Results (Lines 132 - 134):* Compared to the asymmetry indices for sulcal depth and surface area, cortical thickness asymmetry indices were smaller in magnitude (Figure 2).

Functional asymmetries:

- *Results (Lines 151 - 154):* The asymmetry indices of these networks were all within approximately the same range, and were overall similar to the asymmetry indices for sulcal depth and surface area (Figure 2)

We also discuss differences in the magnitude of asymmetry in indices between cortical thickness and other cortical measures investigated here:

- *Discussion (Lines 553 - 557):* The similarity of sulcal depth and surface area asymmetries between neonates and adults suggest that these are broadly conserved, and may explain why the asymmetry indices of these measures are higher than the asymmetry indices of cortical thickness seen here and elsewhere (47, 48).

18. **Comment:** As the paper is focused on asymmetry in the developing brain, it would seem that an asymmetry versus age analysis would be appropriate given they have a decent age span in both the full-term and pre-term cohorts – that would make a nice results Figure 2. As is, one does not get any impression of how asymmetry changes with age in the neonates.

Response: Thank you for this suggestion. After re-running our analyses investigating the effect of PMA on cortical asymmetries, we found that surface area asymmetry in the anterior insula decreased linearly with PMA, and that surface area asymmetry in the mid-insula increasing with PMA.

- *Results (Lines 188 - 192):* Associations between PMA and cortical asymmetries were minimal, with surface area asymmetry in the anterior insula decreasing linearly with PMA, and surface area of a smaller region in the mid-insula increasing linearly with PMA (Figure S3)
- *Discussion (Lines 456 - 467):* Associations between PMA and cortical asymmetries were minimal, with surface area asymmetry in the anterior insula decreasing with age, and surface area asymmetry in the mid-insula increasing with age. The relative lack of age-related associations with cortical asymmetries is consistent with previous studies (17, 23), but also striking given the highly dynamic nature of cortical development over this period. One limitation of the current analysis is that it only tests for linear associations. As non-linear age associations with cortical asymmetries have been recently demonstrated from 4 - 89 years of age (47), future work should investigate whether similar trends exist in the perinatal period.

We also describe the literature investigating the effects of PMA of cortical asymmetries in neonates:

- Introduction (Lines 66 - 73): Relationships between postmenstrual age (PMA) and cortical asymmetries in neonates have received even less attention, with Dubois et al. reporting no association between PMA and sulcal depth asymmetry in preterm neonates born between 26 - 36 weeks PMA (17). To our knowledge, no studies have investigated how biological sex or PMA are associated with functional asymmetry in the neonatal period.
19. **Comment:** Similar to the above comment, it is difficult to pull out how the whole brain asymmetry patterns actually compare to similar studies in older children, adolescents, young adults and older. For a paper with 114 citations, it seems surprising that they don't cite large sample papers that have actually examined cortical thickness asymmetry of whole brain across the lifespan from childhood to old age: e.g. Zhou D et al Neuroimage 2013 – Cortical thickness asymmetry from childhood to older adulthood - n=274, cross-sectional, 5-59 years; Li G et al J Neuroscience 2015 – Spatial patterns, longitudinal development and hemispheric asymmetries of cortical thickness in infants from birth to 2 years of age – n=73, longitudinal 0-2 years (Note: this paper is Ref 19 but is referred to in the Discussion for only one brain region, the middle superior temporal sulcus for some reason – what about the rest of the brain?); Plessen K et al J Neuroscience 2014 – Sex, age, and cognitive correlates of asymmetries in thickness of the cortical mantle across the lifespan – n=215, 7-59 years; etc. It would be useful to have more direct discussions on left/right asymmetry seen in the neonates with this and other similar studies. The Discussion, paragraph 2 seems to be the only paragraph on these MRI study comparisons, but it seems rather limited with a main focus on middle superior temporal sulcus and very little on other brain regions, only one reference by Sha ref 46 (now in PNAS).

Response: The authors thank the reviewer for their comment and suggestions. A similar comment (#13) was made by reviewer #3. We have performed further experiments to directly assess changes in cortical asymmetry from birth to adulthood. In summary, we have:

- Performed vertex-wise analysis of structural asymmetries in 1110 adults from the HCP-YA, after re-running surface registration to improve alignment of cortical folds between hemispheres and across subjects.
- Qualitatively compared the structural asymmetries seen in the HCP-YA with average asymmetry measures calculated in the UKB (previously published by Sha et al. (2021). PNAS, and made publicly available).
- Quantitatively compared structural asymmetries in term neonates against those observed in 1110 young adults from the HCP-YA using template-to-template registration

We have also expanded our introduction where we summarise the relevant previous literature:

- *Introduction (Lines 33 - 56):* For example, one of the most well-characterised structural asymmetries of the human cerebral cortex is the right-deeper-than-left asymmetry of the superior temporal sulcus, which has been demonstrated in fetuses (15, 16), preterm (17, 18) and term neonates (19, 20), children and adults (19, 21, 22). Other sulcal depth asymmetries that have been described include the rightward asymmetry of the parieto-occipital sulcus in neonates (20), and adults (19), and the leftward asymmetry of the lateral fissure (18–20).

In contrast, asymmetries of surface area and cortical thickness during the neonatal period have received less attention. Li et al. demonstrated leftward surface area asymmetries in the precentral, postcentral, middle frontal and cingulate gyri, temporal pole and insula; as well as rightward surface area asymmetry in the inferior parietal, posterior temporal and lateral occipital cortices (20); and rightward cortical thickness

asymmetries in the temporal and lateral occipital cortices, and insula (23). Functional asymmetries, measured using task-based functional MRI (fMRI), are also present at birth and during infancy, and include lateralised responses to speech (24, 25) and music (26).

We also discuss our results in the context of other studies suggested by the reviewer:

- *Discussion (Lines 330 - 339)*: Some of these newly recognised asymmetries have been described in older cohorts. For instance, Zhou et al. demonstrated rightward asymmetry of cortical thickness in the inferior frontal gyrus and medial occipital lobe, and leftward asymmetry of cortical thickness in the supramarginal and angular gyri, and lateral occipital lobe in children and adolescents (46). More recently, Roe et al. demonstrated rightward surface area asymmetry of the medial occipital lobe and caudal anterior cingulate gyrus from as early as 4 years of age (47).
- *Discussion (Lines 525 - 534)*: Unlike sulcal depth, the direction of surface area and cortical thickness asymmetries significantly differed between birth and adulthood. These asymmetry reversals might represent neurodevelopmental milestones. Shaw et al. demonstrated that the pattern of cortical thickness asymmetries observed in the orbitofrontal (left > right) and lateral occipital (right > left) cortices during childhood reversed by adulthood. Moreover, the reversal in orbitofrontal cortical thickness asymmetry between childhood and adulthood was disrupted in those with attention-deficit hyperactivity disorder (104).

20. **Comment:** The conclusion seems to be stepping outside the bounds of the results shown in the paper. Would it not be better to focus on the relationship of asymmetries between structure and function in neonates and how this all compares to older healthy participant asymmetries in cortical structure and resting state function?

Response: Thank you for the comment. The authors agree and the conclusion has been updated.

- *Conclusion (Lines 601 - 612)*: In conclusion, many cortical asymmetries seen later in life are already present at birth, and are minimally associated with preterm birth. Moreover, associations between biological sex and cortical asymmetries only appear to emerge after the neonatal period. Marked differences in cortical thickness asymmetries between neonates and young adults may represent developmental milestones that occur through experience-dependent mechanisms such as intracortical myelination. However, as our results are based on two cross-sectional cohorts, longitudinal studies across the lifespan, beginning in the perinatal period, are required to fully address these hypotheses.

Decision Letter, first revision:

6th October 2022

Dear Dr Williams,

Thank you once again for your revised manuscript, entitled "Structural and functional asymmetry of the neonatal cerebral cortex," and for your patience during the re-review process. I apologize again for the delay.

Your manuscript has now been evaluated by Reviewers #1, #2 and #3. All reviewer feedback is included at the end of this letter. You will see that the reviewers found your manuscript to have improved during revision, but there are a couple of outstanding concerns to address, in particular Referee #2's first point about the accuracy of the registration, and Referee #3's point about the sulcal asymmetry striping pattern being potentially an artifact.

Your revised manuscript must comply fully with our editorial policies and formatting requirements.

In sum, we invite you to revise your manuscript taking into account all reviewer and editor comments. We are committed to providing a fair and constructive peer-review process. Do not hesitate to contact us if there are specific requests from the reviewers that you believe are technically impossible or unlikely to yield a meaningful outcome.

We hope to receive your revised manuscript within 4-8 weeks. I would be grateful if you could contact us as soon as possible if you foresee difficulties with meeting this target resubmission date.

- Include a "Response to the editors and reviewers" document detailing, point-by-point, how you addressed each editor and referee comment. If no action was taken to address a point, you must provide a compelling argument. This response will be used by the editors and reviewers to evaluate your revision.
- Highlight all changes made to your manuscript or provide us with a version that tracks changes.

[REDACTED]

We look forward to seeing the revised manuscript and thank you for the opportunity to review your work. Please do not hesitate to contact me if you have any questions or would like to discuss these revisions further.

Sincerely,

Jamie

Dr Jamie Horder
Senior Editor
Nature Human Behaviour

REVIEWER COMMENTS:

Reviewer #1:

Remarks to the Author:

The authors have addressed my comments – congratulations on an excellent study and paper.

Reviewer #2:

Remarks to the Author:

Thank you for the very deep revision of this interesting paper addressing the evolution of brain asymmetry over development.

there are only a few questions remaining:

1) When registering the neonatal data to the young adult space, how well was the registration done. Was the final registration also manually corrected or did the automatic procedure yield accurate registration?

2) Fig 2 & Fig 3, I do not see many differences and especially in Fig 3, I do not see any region having high blue or yellow (ends of colorbars) values meaning moderate (but within the $p < 0.05$) effect...Is it correct to say that these asymmetries are not very pronounced? Except maybe the left temporo-parietal well known asymmetry.

Reviewer #3:

Remarks to the Author:

I thank the authors for their substantial revisions. The symmetrization of the sulcal depth template and the comparison to the Human Connectome Project Young Adult and UK Biobank are valuable additions in particular.

I have a question regarding interpretation. In the sulcal asymmetry results from the dHCP (Fig. 2, structural) and HCP-YA (Fig. 4a) a centre-surround striping pattern is apparent, with leftwards (red) bands surrounded by rightwards ones (blue). This is particularly salient around the supramarginal gyrus but also present in other areas. What is the origin of this? Is it a genuine property of cortex due to the process of gyrification (differential expansion or axonal tension?). Or an expected property of the sulcal depth extraction method? Or a possible artifact of registration? Some understanding of this would strengthen confidence in the measure and some discussion would be of interest to readers.

It would be helpful if the authors could share as much code as possible on GitHub or a similar platform, to provide a more precise description of what was done and to help others also work with these open datasets.

Author Rebuttal, first revision:**Reviewer #1:**

Remarks to the Author:

The authors have addressed my comments – congratulations on an excellent study and paper.

Reviewer #2:

Remarks to the Author:

Thank you for the very deep revision of this interesting paper addressing the evolution of brain asymmetry over development.

there are only a few questions remaining:

1. When registering the neonatal data to the young adult space, how well was the registration done. Was the final registration also manually corrected or did the automatic procedure yield accurate registration?

We thank the reviewer for their question, since the quality of registration between neonatal space and adult space is critical for comparing asymmetries between these two age groups. As described in the methods, registration of asymmetry maps to HCP space was achieved through template-to-template registration.

- *Methods (lines 816 - 823)*: Comparison of structural cortical asymmetries between birth and adulthood was achieved by registering the symmetric dHCP 40-week PMA sulcal depth template to the symmetric HCP fs_LR 32k sulcal depth template using MSM (52, 53), creating a template-to-template registration deformation. This deformation was then used to resample single subject structural cortical asymmetry maps from neonatal to adult space.

Rigorous visual QC of the dHCP native-to-template registration, and of dHCP-to-HCP template registration, was performed prior to analysis. We have attached a figure below to highlight registration quality.

(a) HCP MSMSulc template. (b) dhcpSym 40 week PMA template registered to HCP MSMSulc template. (c) single HCP subject in HCP MSMSulc space. (d) single dHCP subject in HCP MSMSulc space. Dots in a - d represent outline of sulci in HCP MSMSulc template in (a).

Overall, the authors feel that image registration using folding-based alignment is accurate, and have added this to the methods:

- Methods (lines 847 - 848): Quality control for all surface registrations was performed by visual inspection, and were deemed accurate.
- Methods (lines 856 - 859): The quality of the dHCP-to-HCP template registration was visually inspected and deemed to be as accurate as individual subject-to-template registrations.

2. Fig 2 & Fig 3, I do not see many differences and especially in Fig 3, I do not see any region having high blue or yellow (ends of colorbars) values meaning moderate (but within the $p < 0.05$) effect...Is it correct to say that these asymmetries are not very pronounced? Except maybe the left temporo-parietal well known asymmetry.

We agree that the asymmetry indices are small, although statistically significant. For the structural asymmetries presented in figures 2 and 3, the asymmetry indices are approximately in the same range as the structural asymmetry indices for the HCP-YA experiments, and the UK biobank asymmetry indices presented in figure 4. The

asymmetry indices reported here are similar to those reported in the literature, for example:

- Zhou et al. (2006). Cortical thickness asymmetry from childhood to older adulthood. *NeuroImage*
- Kong et al. (2018). Mapping cortical brain asymmetry in 17,141 healthy individuals worldwide via the ENIGMA consortium. *PNAS*
- Sha et al. (2021). Handedness and its genetic influences are associated with structural asymmetries of the cerebral cortex in 31,864 individuals. *PNAS*
- Williams et al. (2022). Comparing brain asymmetries independently of brain size. *NeuroImage*

Reviewer #3:

Remarks to the Author:

I thank the authors for their substantial revisions. The symmetrization of the sulcal depth template and the comparison to the Human Connectome Project Young Adult and UK Biobank are valuable additions in particular.

1. I have a question regarding interpretation. In the sulcal asymmetry results from the dHCP (Fig. 2, structural) and HCP-YA (Fig. 4a) a centre-surround striping pattern is apparent, with leftwards (red) bands surrounded by rightwards ones (blue). This is particularly salient around the supramarginal gyrus but also present in other areas. What is the origin of this? Is it a genuine property of cortex due to the process of gyrification (differential expansion or axonal tension?). Or an expected property of the sulcal depth extraction method? Or a possible artifact of registration? Some understanding of this would strengthen confidence in the measure and some discussion would be of interest to readers.

We thank the reviewer for their astute observation regarding the appearance of the sulcal depth asymmetry index maps. It is difficult to fully disentangle the causes of this phenomenon but it is likely to be a combination of true differences in sulcal depth asymmetry and unavoidable residual misalignment, as shown in the figure below.

Figure (a). Schematic of left and right cortical folds in two registration scenarios: **1. Perfect alignment**, where the location of the gyral crests/sulcal fundi/inflection points (transition between gyrus and sulcus) correspond across left and right hemispheres. In this instance, the left cortical fold has higher amplitude (greater gyral height and lower sulcal depth), which creates a centre of leftward asymmetry flanked by areas of rightward asymmetry. In the other scenario, **2. Residual misalignment**, the same centre surround pattern has been produced despite there being no true differences in sulcal depth asymmetry.

Despite driving image registration to a perfectly symmetric template, it is not possible for all subjects to have *exact* anatomical correspondence across the left and right hemispheres, given the morphological variability of the cortex. In addition to this, image registration is an ill-posed problem with many possible solutions, where regularisations are introduced to enforce biologically 'plausible' solutions. It is possible that the effect of regularisation is different for the left and right hemispheres. Unfortunately, we are not aware of any solutions to this problem.

Given the above, it is not possible to tell with certainty which patterns are produced by which phenomenon. However, very few of these centre-surround patterns are statistically significant, which is suggestive that the patterns observed in figures 2 and 4 should largely be considered background noise. .

We have updated the results and discussion accordingly:

- Results (lines 115 - 120): The sulcal depth asymmetry index map demonstrated an interesting centre-surround pattern, where regions of asymmetry in one direction were flanked by regions of opposing asymmetry, for example a centre of rightward asymmetry in the supramarginal gyrus surrounded by regions of leftward asymmetry.
- Results (lines 241 - 243): The HCP-YA sulcal depth asymmetry index map also demonstrated the same centre-surround pattern.
- Discussion (lines 585 - 609): The sulcal depth asymmetry index maps across both dHCP (Figure 2) and HCP-YA (Figure 4a) cohorts demonstrated a

marked centre-surround pattern. It is difficult to fully disentangle the causes of this phenomenon but it is likely to be a combination of true differences in sulcal depth asymmetry and unavoidable residual image registration misalignment. Despite driving registration to a template that is perfectly symmetric (Figure 6), it is not possible for a template to capture all possible variations in cortical folding (116–118). Moreover, asymmetric presence of sulci has been demonstrated in a number of cortical regions including the (para)cingulate cortex (119, 120) and the frontal operculum (121), suggesting that in some instances folding correspondence between hemispheres within subjects is also not possible. This anatomical variability is compounded by the fact that image registration is an ill-posed problem with many possible solutions, where regularisations are introduced to enforce solutions that are considered biologically 'plausible'. It is possible that the effect of regularisation is different for the left and right hemispheres. Given the above, it is not possible to tell with certainty which centre-surround patterns are true biological differences or a product of residual misalignment. Regardless, very few of these patterns were statistically significant, and for the purpose of the analyses presented here, should be seen as background noise.

- Supplementary results (figure S13, page 14 of 15): Schematic of left and right cortical folds in two registration scenarios: 1. Perfect alignment, where the location of the gyral crests/sulcal fundi/inflection points (transition between gyrus and sulcus) correspond across left and right hemispheres. In this instance, the left cortical fold has higher amplitude (greater gyral height and lower sulcal depth), which creates a centre of leftward asymmetry flanked by areas of rightward asymmetry. In the other scenario, 2. Residual misalignment, the same centre surround pattern has been produced despite there being no true differences in sulcal depth asymmetry.
2. It would be helpful if the authors could share as much code as possible on GitHub or a similar platform, to provide a more precise description of what was done and to help others also work with these open datasets.

The authors have added the code used for image processing and analysis at <https://github.com/metrics-lab/CorticalAsymmetry>. This has been updated in the main

text:

- Code availability (lines 1001 - 1002): Code used to perform image processing and asymmetry analyses is available at <https://github.com/metrics-lab/CorticalAsymmetry>.

Decision Letter, second revision:

11th January 2023

Dear Dr. Williams,

Thank you for submitting your revised manuscript "Structural and functional asymmetry of the neonatal cerebral cortex" (NATHUMBEHAV-211016953B). It has now been seen by the original referees and their comments are below.

As you can see, the reviewers find that the paper has improved in revision. We will therefore be happy in principle to publish it in Nature Human Behaviour, pending minor revisions to satisfy the referees' final requests and to comply with our editorial and formatting guidelines.

We are now performing detailed checks on your paper and will send you a checklist detailing our editorial and formatting requirements within a week. Please do not upload the final materials and make any revisions until you receive this additional information from us.

Sincerely,

Jamie

Dr Jamie Horder
Senior Editor
Nature Human Behaviour

Reviewer #2 (Remarks to the Author):

The authors have now sufficiently addressed my questions and have outlined the potential limitations of the results due to inherent registration issues

Thank you for reviewing this interesting imaging based study on newborn brain development

Final Decision Letter:

Dear Dr Williams,

We are pleased to inform you that your Article "Structural and functional asymmetry of the neonatal cerebral cortex", has now been accepted for publication in *Nature Human Behaviour*.

Please note that *Nature Human Behaviour* is a Transformative Journal (TJ). Authors whose manuscript was submitted on or after January 1st, 2021, may publish their research with us through the traditional subscription access route or make their paper immediately open access through payment of an article-processing charge (APC). Authors will not be required to make a final decision about access to their article until it has been accepted. IMPORTANT NOTE: Articles submitted before January 1st, 2021, are not eligible for Open Access publication. Find out more about Transformative Journals

An online order form for reprints of your paper is available at <https://www.nature.com/reprints/author-reprints.html>. All co-authors, authors' institutions and authors'

funding agencies can order reprints using the form appropriate to their geographical region.

With best regards,

Jamie

Dr Jamie Horder

Senior Editor
Nature Human Behaviour